# Real-time self-supervised denoising for high-speed fluorescence neural imaging

Yiqun Wang ●[1,7], Yuanjie Gu ●[1,7], Jianping Wang[1], Ang Xuan[1], Cihang Kong[2], Wei-Qun Fang[3], Dongyu Li ●[4], Dan Zhu ●[5], Fengfei Ding ●[6] ✉ & Biqin Dong ●[1] ✉

Self-supervised denoising methods significantly enhance the signal-to-noise ratio in fluorescence neural imaging, yet real-time solutions remain scarce in high-speed applications. Here, we present the FrAme-multiplexed Spatio-Temporal learning strategy (FAST), a deep-learning framework designed for high-speed fluorescence neural imaging, including in vivo calcium, voltage, and volumetric time-lapse imaging. FAST balances spatial and temporal redundancy across neighboring pixels, preserving structural fidelity while preventing over-smoothing of rapidly evolving fluorescence signals. Utilizing an ultra-light convolutional neural network, FAST enables real-time processing at speeds exceeding 1000 frames per second, substantially surpassing the acquisition rates of most high-speed imaging systems. We also introduce an intuitive graphical user interface that integrates FAST into standard imaging workflows, providing a real-time denoising tool for recorded neural activity and enabling downstream analysis in neuroscience research that requires millisecond-scale temporal precision, particularly in closed-loop studies.

Intravital neural imaging is critical for understanding brain science, especially in examining neural circuit dynamics underlying behavioral paradigms and investigating disease-specific mechanisms such as neurodegeneration and epilepsy[1–6]. Fluorescence imaging, an essential technique in neuroscience to characterize neural structures and activities, is currently facing technical ceilings due to the limitations in fluorescent probes and light scattering in deep brain tissues[7–10]. Increases in imaging depth and speed often come at the expense of signal-to-noise ratio (SNR), posing challenges for high-quality data acquisition. Recent advances in denoising methods aim to address these issues[11–15], yet different imaging scenarios demand tailored denoising solutions[15–18]. For example, structural imaging emphasizes preserving high spatial resolution[19–22], while functional imaging prioritizes high temporal resolution[14–17]. Furthermore, in closed-loop neural

modulation, the process must seamlessly integrate imaging, analysis, and stimulation[23], adding further complexity to the already demanding requirements for high-speed, high-SNR imaging. Therefore, an ideal denoising tool that is universally applicable across different imaging scenarios and enables real-time feedback remains highly desired but is still lacking, particularly in ultra-high-speed neural activity recording at the millisecond scale.

Imaging denoising has emerged as an enabling technology to reconcile the fundamental constraints of neural imaging systems and break the inherent trilemma between spatial resolution, temporal sampling rate, and SNR. Since the advent of deep learning-based denoising methods, self-supervised approaches[24–29] have emerged as powerful tools for image enhancement in neural imaging due to their exceptional denoising performance. Self-supervised methods

[1]College of Biomedical Engineering, Yiwu Research Institute, Fudan University, Shanghai, China. [2]Institute for Translational Brain Research, Fudan University, Shanghai, China. [3]Songjiang Hospital and Songjiang Research Institute, Shanghai Key Laboratory of Emotions and Affective Disorders, Shanghai Jiao Tong University School of Medicine, Shanghai, China. [4]School of Optical Electronic Information-Advanced Biomedical Imaging Facility, Huazhong University of Science and Technology, Wuhan, Hubei, China. [5]Britton Chance Center for Biomedical Photonics, MoE Key Laboratory for Biomedical Photonics-Wuhan National Laboratory for Optoelectronics, Advanced Biomedical Imaging Facility, Huazhong University of Science and Technology, Wuhan, Hubei, China. [6]Department of Pharmacology, School of Basic Medical Sciences, Fudan University, Shanghai, China. [7]These authors contributed equally: Yiqun Wang, Yuanjie Gu. ✉e-mail: fengfei_ding@fudan.edu.cn; dongbq@fudan.edu.cn

performed without ground truth, leveraging inherent spatiotemporal redundancies, such as repetitive patterns or correlations in neural imaging datasets, to extract clean image features from noisy inputs. The existing self-supervised denoising methods fall into two main categories: adjacency-based methods and blind-spot-based methods. Adjacency-based approaches[17,22,25,26] resample time intervals between frames in multi-frame datasets or spatial pixel intervals in single-frame images, leveraging temporal or spatial redundancy. They require high-resolution imaging to prevent over-smoothing and artifacts. Blind-spot-based methods[14–16], degrade the central frame and reconstruct it using unmasked pixels from adjacent frames. These methods are hindered by long training and testing times, often posing challenges for real-time processing.

Here, we present the FrAme-multiplexed SpatioTemporal learning strategy (FAST), enabling real-time, self-supervised denoising across diverse neural imaging scenarios. FAST is carried out within an ultra-lightweight 2D convolutional network containing only 0.013 M parameters. This lightweight architecture significantly enhances computational efficiency, enabling real-time processing even on resource-limited hardware. The model processes full-size, multi-frame images without partitioning, breaking through the denoising speed ceiling of 1000 frames per second (FPS) and achieving an unprecedented speed of up to 2100 FPS. This approach differs from existing real-time denoising methods, which typically depend on fixed temporal sampling and complex 3D network architectures to capture spatiotemporal information. FAST, by introducing an adaptive frame-multiplexed spatiotemporal sampling strategy, enables the use of a lightweight 2D network without sacrificing denoising performance. This design allows the temporal window to be flexibly adjusted according to the signal dynamics, facilitating a tailored balance between spatial and temporal information. Consequently, FAST can effectively reduce artifacts and over-smoothing, particularly in scenarios involving rapid or non-stationary neural activity. By decoupling the reliance on heavy network structures and instead leveraging more informative sampling, FAST extends the applicability of real-time self-supervised denoising to experimental scenarios that have previously posed significant challenges for conventional approaches.

In this work, we validated FAST's high-performance denoising and real-time capabilities across three in vivo neural fluorescence imaging scenarios: calcium imaging, voltage imaging, and volumetric time-lapse imaging, using models such as neurons in mice and zebrafish, as well as astrocytes in mice. We also developed a graphical user interface (GUI) for FAST, enabling users to easily train custom denoising models on their data and perform real-time inference. Our results demonstrate that FAST enhances cellular morphology restoration, improves neuron segmentation, and increases the accuracy of 3D calcium event extraction in astrocytes. For voltage imaging, comparisons with synchronized electrophysiological recordings reveal that our method preserves spike shapes and significantly boosts the correlation between voltage transients and electrophysiological recordings. By enabling real-time, high-performance denoising, FAST has the potential to transform neural imaging workflows, facilitating more accurate and efficient studies of brain function and disease mechanisms.

## Results

### Principle and performance of FAST
FAST employs a lightweight and efficient network architecture (Supplementary Fig. 3 and details in Methods) with significantly fewer parameters compared to other deep learning-based denoising models (Fig. 1c). Specifically, its parameter count is 8 to 36 times lower than methods like DeepCAD-RT[17], SRDTrans[22], DeepVid[15], and SUPPORT[16], which rely on more complex structures such as 3D convolutional networks[30], Swin Transformer[31], ResNet[32] architectures, or ensemble networks. This substantial reduction in parameters not only minimizes memory usage and computational overhead but also directly enhances

FAST's processing speed, making it highly efficient and well-suited for real-time applications. For instance, benchmark tests using input image sequences with dimensions $512 \times 192 \times 5000$ (x-y-t) on an NVIDIA RTX A6000 GPU (Fig. 1d) demonstrated that FAST achieved processing speeds orders of magnitude exceeding those of other real-time methods under consistent experimental conditions. Specifically, FAST's processing speed was 18 to 2559 times faster than competing methods, surpassing the 1000 FPS threshold and reaching an impressive 1100.45 FPS. These results highlight its unparalleled efficiency and suitability for real-time applications. The adjustable trade-off parameters, spatiotemporal sampling strategy, and lightweight network architecture integrated into FAST enable high PSNR and SSIM values while maintaining processing speeds beyond real-time across various imaging scenarios (Supplementary Sections 3-7 and Supplementary Figs. 4-8).

FAST is a real-time denoising method that integrates an efficient processing pipeline and a tailored training framework to achieve high-speed, high-quality denoising across diverse imaging scenarios. Its real-time denoising pipeline is designed to process imaging data in a synchronized and efficient manner (Fig. 1e). During imaging, acquired frames are temporarily stored in batches within a solid-state drive (SSD) buffer. These frames are read into a noisy queue and processed by the trained FAST model in a first-in, first-out (FIFO) manner. The denoised frames are subsequently stored in a denoised queue, enabling synchronized display of both pre- and post-denoised frames. This pipeline not only supports real-time visualization but also enables customizable online analysis or processing, providing flexibility for diverse imaging needs. The real-time denoising pipeline operates through three parallel threads for image acquisition, denoising, and display, all coordinated by the FAST GUI (Fig. 1f, Supplementary Fig. 9 and details in Methods), which ensures synchronized operations and user-friendly control. This threaded workflow ensures seamless and synchronized online processing, enabling FAST to handle continuous data acquisition and processing efficiently over time. Furthermore, the pipeline is highly versatile, supporting various imaging samples and accommodating a wide range of imaging speeds for both 2D and 3D imaging modalities (Fig. 1g).

The system's real-time performance and responsiveness are further demonstrated in Supplementary Video 1 and Supplementary Section 8, where a live in vivo two-photon calcium imaging experiment is presented. In this demonstration, both raw and denoised images are displayed simultaneously at an acquisition rate of 30 Hz, providing direct visual evidence of FAST's capability for real-time denoising under practical experimental conditions.

### Rapid processing of intracellular Ca$^{2+}$ imaging data from neuronal populations with FAST
Fluorescent genetically encoded calcium indicators and two-photon microscopy enable the simultaneous capture of the functional activity of neurons. However, the segmentation of neuronal somata is often hampered by imaging signal noise, which affects both the number of neurons detected and the accuracy of their contours[33].

To address these issues, we applied FAST to denoise calcium imaging data acquired from the mouse vS1 region[34]. The experimental pipeline involved denoising raw GCaMP6s calcium imaging videos using five different methods (DeepCAD-RT, SRDTrans, DeepVid, SUP-PORT, and FAST), followed by temporal maximum intensity projection (MIP) and segmentation using Cellpose[35] (Fig. 2a). The denoising performance was evaluated through qualitative analysis of neuronal morphology and quantitative metrics, including segmentation accuracy, recall, and F1 score, using manually annotated ground truth (see details in Methods). Among the five denoising methods tested, FAST demonstrated the most consistent performance in preserving neuronal structures and improving segmentation outcomes. Figure 2b illustrates the raw calcium imaging data and the corresponding

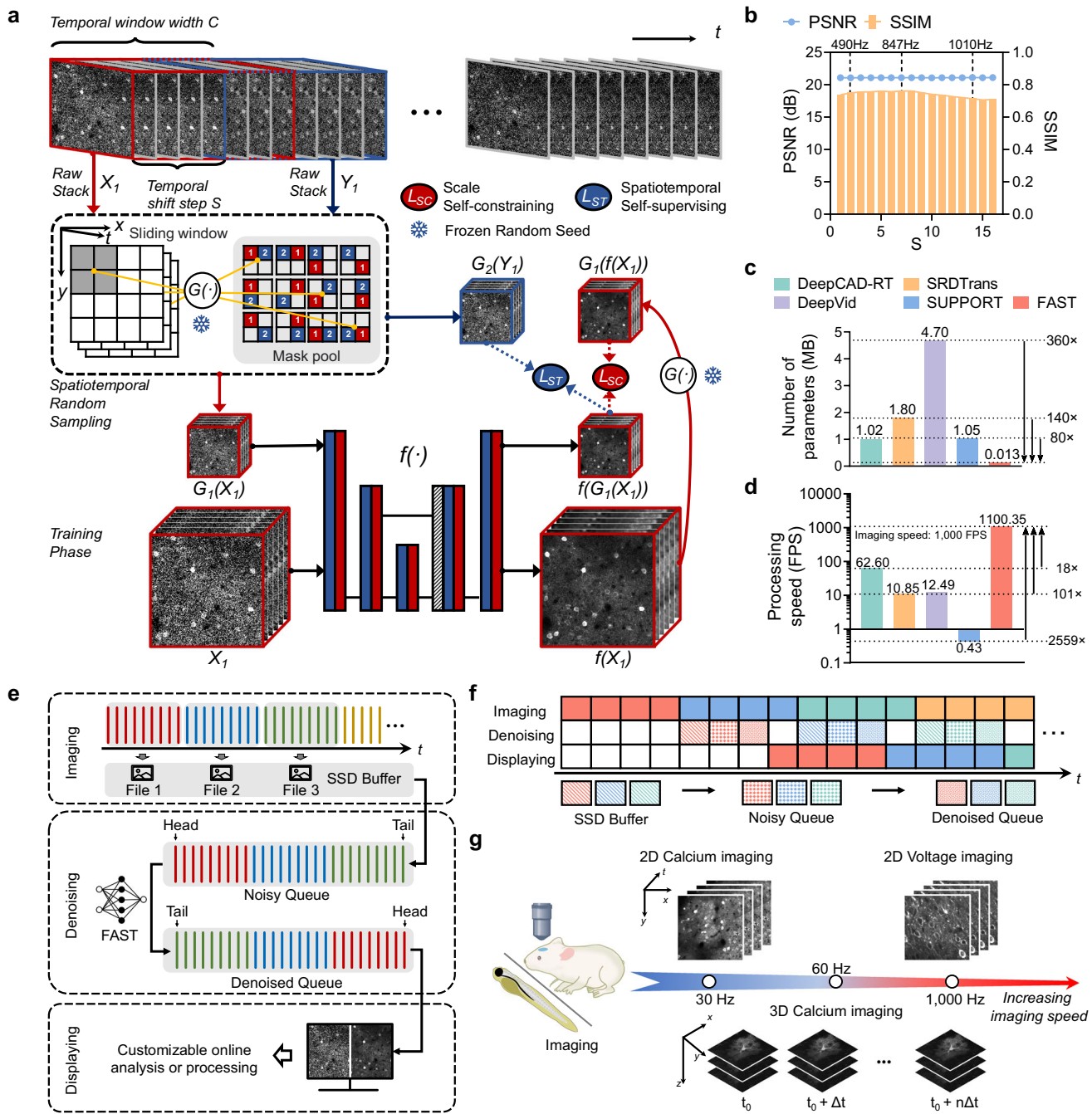

**Fig. 1 | Principle and performance of FAST. a** Training framework of FAST: The time-lapse raw stack acquired during imaging is divided into the network's input and target through spatiotemporal random sampling. For example, $X_1$ and $Y_1$ represent a pair of images obtained after the first temporal subsampling of the raw stack. Temporal subsampling is achieved by sliding a window with a width $C$ (temporal window width) and a shift step size $S$ (temporal shift step) along the time axis. Here, $C$ and $S$ serve as trade-off factors, balancing temporal resolution and processing speed for optimal denoising performance. Spatial subsampling, denoted as $G(\cdot)$, randomly divides neighboring pixels in space. All possible spatial division patterns are shown in the mask pool. The snowflake symbol indicates that the random seed for spatiotemporal subsampling is fixed. This ensures consistent spatial adjacency within each sample pair, while allowing relationships between different pairs to vary. $G_1(\cdot)$ and $G_2(\cdot)$ represent the results obtained through spatiotemporal subsampling, where their pixels satisfy spatial adjacency relationships. The parameters of the denoising network $f(\cdot)$ are optimized using scale self-constraining ($L_{SC}$) and spatiotemporal self-supervising ($L_{ST}$). **b** Impact of stride S on

FAST's performance using simulated data. Blue dots represent the peak signal-to-noise ratio (PSNR) and orange boxes show the structural similarity index (SSIM). Processing speeds are indicated for three S values. The noisy input has a PSNR of 12.34 dB and an SSIM of 0.03. **c** FAST has substantially fewer parameters than other deep learning models, reducing memory and computational requirements. **d** For input image sequences with dimensions 512×192×5000 (x-y-t), processing speeds are measured on an NVIDIA RTX A6000 GPU and reported in frames per second. Under consistent experimental conditions, FAST achieves a processing speed over 80 times faster than other real-time methods[16]. **e, f** Real-time, multi-threaded denoising pipeline. Three parallel threads manage image acquisition, denoising, and display. Acquired frames are buffered, processed by the trained FAST network in a first-in, first-out (FIFO) queue, and then displayed alongside the raw data for synchronized comparison and optional online analysis. **g** The pipeline is adaptable to various imaging samples, speeds, and supports both 2D and 3D time-lapse imaging.

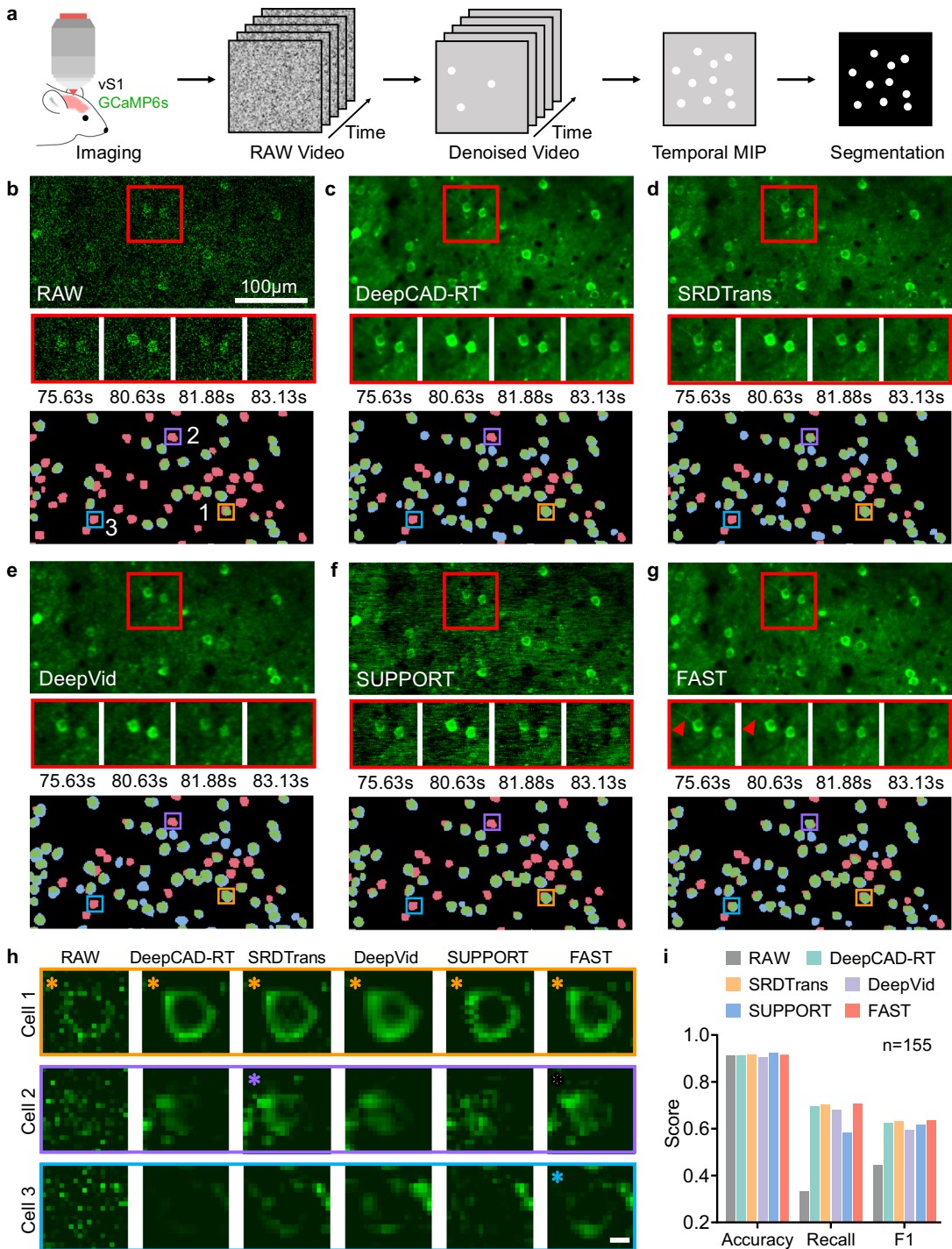

segmentation results. The raw data were heavily contaminated with noise, and distorted neuronal contours, leading to a high rate of false negatives (red regions) during segmentation. For example, in the raw data, Cellpose failed to detect ~65% of neurons (100 out of 155). This resulted in significant information loss.

After denoising, FAST demonstrated excellent performance in restoring neuronal morphology and improving segmentation outcomes (Fig. 2c–g, Supplementary Section 9 and Supplementary Video 2). In the magnified view of Fig. 2g, arrowheads indicate dendrites that were originally obscured by noise in the raw data. After denoising, these structures were revealed with notably improved clarity and continuity by FAST. We further compared the raw and denoised images in Fig. 2h, focusing on the segmentation outcomes

for three representative neurons (Cells 1–3) to illustrate the challenges of segmentation under noisy conditions. For Cell 1, all methods enabled successful segmentation by Cellpose, as its contour remained relatively clear and intact across the different denoising results. For Cell 2, only the SRDTrans-denoised and FAST-denoised images allowed Cellpose to segment the neuron, whereas the other methods produced results with insufficiently continuous structures, making segmentation difficult. For Cell 3, the FAST-denoised image exhibited the most complete contour and the clearest boundary, which greatly facilitated accurate segmentation.

The quantitative evaluation confirmed the segmentation performance of FAST (Fig. 2i). FAST achieved the highest Recall (0.71) and F1 score (0.64), indicating its effectiveness in preserving neuronal

**Fig. 2 | FAST enhances Ca²⁺ imaging quality and analytical accuracy. a** Raw GCaMP6s calcium imaging videos from the mouse vS1 region were first denoised using different denoising methods. Subsequently, temporal maximum intensity projection (MIP) was applied to the videos. The resulting projection images were then segmented using Cellpose. **b** In vivo neural population calcium imaging data were obtained from ref. 32 The top panel shows the raw calcium imaging data with a red box indicating a region of interest (ROI). Enlarged images of the ROI at four different time points during a single spike event (~7.5 s time window) are presented, illustrating neuronal activity in various states to evaluate denoising performance. The bottom panel displays the corresponding segmentation results, with manual annotations used as ground truth: correctly segmented regions (true positives) are shown in green, missed regions (false negatives) in red, and extra regions (false positives) in blue. Three neuron regions are highlighted with boxes in the bottom panel, and their corresponding magnified segmentation results are shown in (**h**). Scale bar, 100 μm. **c**–**g** Example denoising and segmentation results from raw data

processed using DeepCAD-RT, SRDTrans, DeepVid, SUPPORT, and FAST. Following the same structure as (**b**). In g, arrowheads indicate the fine dendrites that become observable after FAST denoising. The images shown are representative frames, and similar results were obtained across all 8000 frames analyzed. **h** Example neurons, zoomed in from the boxed regions in (**b**–**g**), illustrating segmentation results across different denoising methods. Each row represents a single neuron, while each column shows the maximum intensity projection of the raw data or data denoised by one of the five methods. To ensure a fair visual comparison, all magnified views shown were cropped from full-sized images that had been globally normalized to a single, unified intensity range, with no subsequent, individual re-normalization applied. In **h**, asterisks indicate neurons that were successfully segmented by Cellpose. Scale bar, 5 μm. **i** Segmentation performance (Accuracy, Recall, and F1 score) was calculated for $n = 155$ neurons, all located within a single imaging dataset.

structures. SUPPORT, despite achieving the highest Accuracy (0.93), showed lower Recall and F1 scores, reflecting a trade-off between precision and recall. Notably, when applying the official SUPPORT implementation to our calcium imaging dataset, we observed the presence of horizontal fringe artifacts. FAST, by comparison, balanced these metrics better, achieving a similar Accuracy (0.92) with higher Recall and F1 scores. These findings suggest that FAST provides a reliable approach for denoising calcium imaging data, which could facilitate more accurate neuronal segmentation and downstream analyses in neuroscience research.

## Rapid enhancement in high-speed voltage imaging with FAST

Voltage imaging is a powerful technique for monitoring rapid neuronal activity, capable of resolving millisecond-scale changes in membrane potential with high temporal precision. Unlike calcium imaging, which reflects indirect activity signals, voltage imaging provides a direct and precise representation of electrophysiological events. Recent advances in voltage indicators, such as QuasAr6a[36] and zArchon1[37], have significantly contributed to enabling imaging speeds up to 1,000 Hz, allowing researchers to resolve fast voltage transients critical for understanding neural dynamics. However, such high-speed imaging significantly reduces photon collection, leading to a low SNR. The resulting noise not only obscures neuronal morphology but also distorts voltage signals, making it challenging to extract meaningful dynamics. These challenges necessitate robust denoising methods that can preserve both the spatial and temporal fidelity of voltage imaging data. Here, we demonstrate that FAST effectively addresses these challenges by removing noise while faithfully preserving voltage transients. We evaluated FAST's performance through simulation experiments, in vivo single-neuron imaging in the mouse cortex, and population-level imaging in the zebrafish spinal cord. Our results show that FAST significantly improves both the clarity of neuronal morphology and the fidelity of extracted voltage signals, enabling more accurate analysis of neural activity under extreme imaging conditions.

To systematically evaluate FAST's ability to recover rapid dynamic signals, we conducted a simulation experiment using synthetic two-photon imaging data (Fig. 3a and Supplementary Fig. 10). Virtual spike signals with varying widths (2 ms, 4 ms, 6 ms, and 8 ms) were generated and assigned to individual neurons. These signals served as noise-free ground truth (GT) data. Mixed Poisson-Gaussian noise was then added to simulate realistic imaging conditions, creating noisy data as input for denoising networks. Five denoising methods, including DeepCAD-RT, SRDTrans, DeepVid, SUPPORT, and FAST, were applied to the noisy data, and their performance was assessed by calculating Pearson correlation coefficients between the denoised ΔF/F traces and the ground truth.

Figure 3b shows the correlation coefficients across different spike widths, while Fig. 3c presents representative ΔF/F traces. FAST consistently achieved the highest correlation coefficients across all spike

widths (Fig. 3b), significantly outperforming other methods, especially for narrower spikes (e.g., 2 ms and 4 ms). Notably, for spike widths below 6 ms, the correlation coefficients of the other five denoising methods were even lower than those of the noisy data, indicating that these five methods failed to recover rapid dynamics and instead treated them as noise. In contrast, FAST reliably preserved the original spike shapes, as illustrated by the representative ΔF/F traces in Fig. 3c.

To validate FAST's performance on real-world data, we applied it to in vivo voltage imaging of single neurons expressing QuasAr6a in layer 2/3 of the mouse cortex (Fig. 3d, Supplementary Section 10 and Supplementary Video 3). Imaging was conducted at 1000 Hz using background-rejection structured illumination fluorescence microscopy[35], alongside simultaneous patch-clamp electrophysiological recordings. The raw imaging data were heavily contaminated with noise, which obscured neuronal boundaries and distorted voltage signals. FAST denoising significantly improved the clarity of neuronal morphology and restored voltage signals, as demonstrated by the ΔF/F traces extracted from 1000 pixels randomly selected from the entire neuronal ROI, including both membrane and cytoplasmic regions (Fig. 3d, right panel). In the enlarged time window shown in Fig. 3e, FAST-denoised traces closely matched the electrophysiological recordings, accurately capturing the timing and amplitude of individual spikes. To quantitatively evaluate FAST's performance, we calculated the Pearson correlation coefficients between the ΔF/F traces and the simultaneously recorded electrophysiological signals (Fig. 3f). FAST significantly increased the correlation coefficients compared to the raw data ($P < 0.001$), indicating a substantial improvement in signal fidelity.

To further evaluate FAST's applicability to population-level imaging, we applied it to in vivo voltage imaging of zebrafish spinal cord neurons expressing the voltage indicator zArchon1. Imaging was performed using light-sheet fluorescence microscopy at 1000 Hz (Fig. 3g, Supplementary Section 11 and Supplementary Video 4). The raw data[16] were severely degraded by noise, making it difficult to delineate neuronal boundaries or extract reliable voltage signals. FAST significantly improved the visibility of neuronal structures, as shown in the manually annotated ROIs (Fig. 3g, right panel). Enlarged views of three representative neurons further illustrate FAST's ability to recover fine morphological details that were otherwise obscured in the raw data. We next analyzed the ΔF/F traces extracted from the three representative neurons (Fig. 3h). In the raw data, the traces were dominated by noise, obscuring the underlying voltage dynamics. After FAST denoising, the traces revealed clear and consistent spike patterns, reflecting the expected neural activity. In the enlarged time window (Fig. 3i), FAST-denoised traces accurately captured the timing and shape of individual spikes, which were indistinguishable in the raw data.

These results demonstrate that FAST effectively addresses the challenges of low SNR in high-speed voltage imaging. By preserving both spatial and temporal fidelity, FAST enables the recovery of rapid

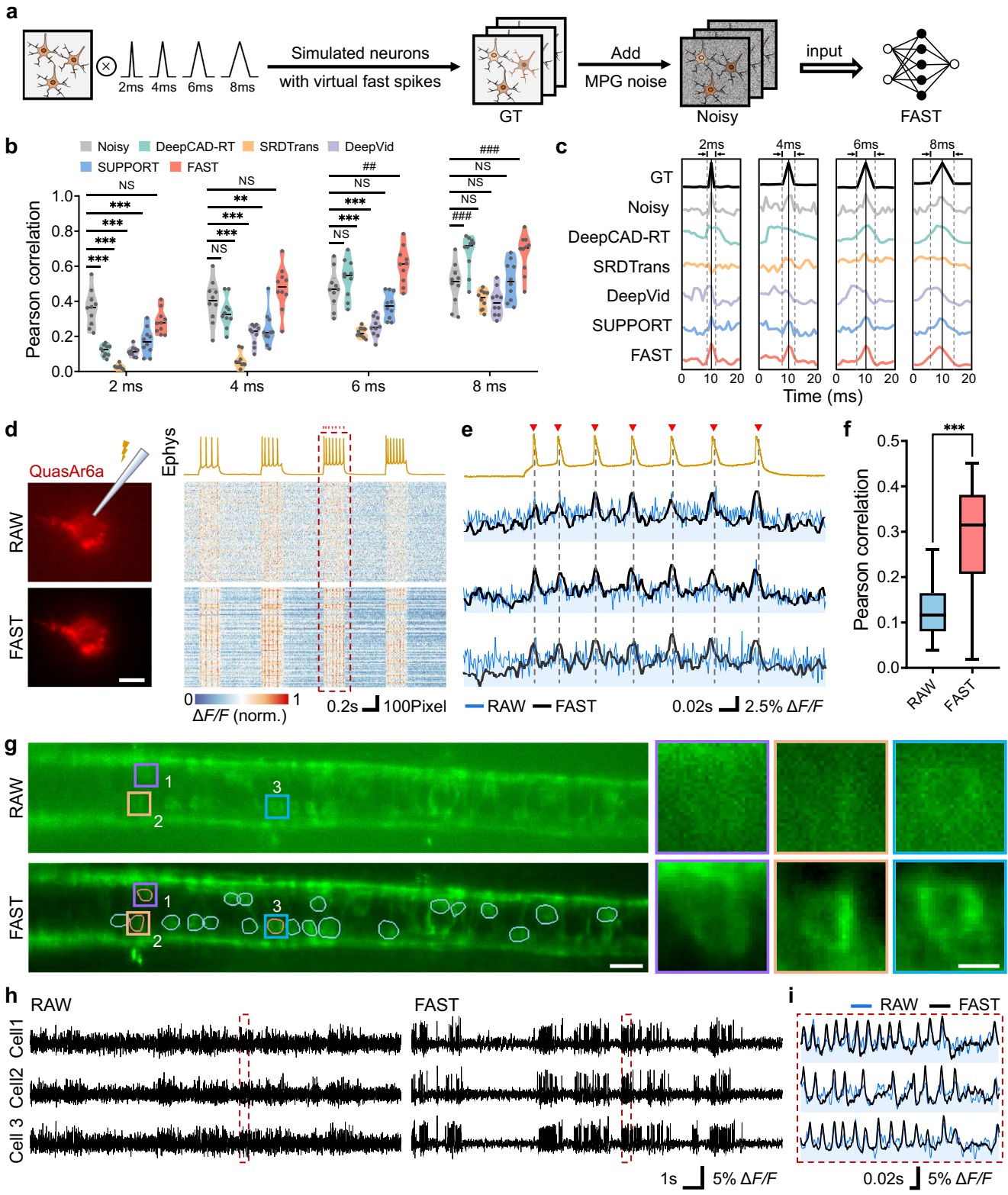

dynamic signals that are otherwise obscured by noise. In simulation experiments, FAST outperformed other methods in restoring spike shapes, particularly for narrower spike widths, where its advantage was most pronounced. In real-world applications, FAST significantly improved the clarity of neuronal morphology and the fidelity of voltage signals in both single-neuron and population-level imaging. These findings highlight FAST's potential as a robust tool for enhancing voltage imaging under challenging conditions, paving the way for more accurate analysis of neural activity.

## Rapid processing of volumetric imaging of cortical astrocytes with FAST

Cortical protoplasmic astrocytes, the most abundant glial cells in the central nervous system, play essential roles in maintaining interstitial homeostasis and regulating neural circuits[38–40]. Their processes closely interact with neuronal synapses, blood vessels, and other cellular structures, integrating into the broader neural network[41–43]. Recent studies[44,45] using volumetric imaging have expanded our understanding of astrocytic intracellular $Ca^{2+}$ dynamics, revealing their

**Fig. 3 | High-fidelity voltage imaging enhancement via FAST. a** Workflow for simulating two-photon voltage imaging data. Ground truth (GT) signals with varying spike widths (2 ms, 4 ms, 6 ms, or 8 ms), were combined with mixed Poisson-Gaussian noise to generate network inputs. **b** Statistical analysis of denoising performance across spike widths. Pearson correlation coefficients were calculated between each neuron's denoised signals (five methods) and the GT signal. Data are presented as violin plots ($n = 10$ neurons). Statistical significance was determined using two-sided two-way ANOVA followed by Tukey's multiple comparisons test. Significance levels are indicated as follows: NS ($P \geq 0.05$); ** ($P < 0.01$, decrease); *** ($P < 0.001$, decrease); ## ($P < 0.01$, increase); ### ($P < 0.001$, increase). **c** Representative $\Delta F/F$ traces from a single simulated neuron, comparing GT, noisy, and denoised signals across the four spike widths. **d** Raw and FAST-denoised images with in vivo simultaneous voltage imaging and electrophysiology. Left: Raw and FAST-denoised images of a QuasAr6a-expressing neuron in mouse cortex (L2/3) at 1000 Hz (data from ref. 53). Right: Electrophysiology trace

(top) and $\Delta F/F$ traces from 1000 pixels within the neuron ROI (annotated using the ROI Manager toolbox in Fiji[54]) for noisy (middle) and FAST-denoised (bottom) data. Red box is magnified in (**e**). Scale bar, 10 μm. Representative frames are shown in the figure, and similar results were obtained across $n = 4982$ frames in all experiments. **e** Enlarged view from (**d**). Overlaid raw and FAST-denoised $\Delta F/F$ traces from three representative pixels are aligned with the electrophysiology trace. Red triangles mark spike peaks. **f** Pearson correlation between the $n = 100$ single-pixel $\Delta F/F$ traces and the electrophysiology recording in (**d**). FAST significantly improves correlation (***$P < 0.001$, unpaired two-tailed t-test). **g** Voltage imaging of zebrafish spinal cord neurons using light sheet microscopy. Raw and FAST-denoised images of zArchon1-expressing neurons imaged at 1000 Hz (data from ref. 16). ROIs for three representative cells are manually annotated and magnified on the right. Scale bars, 20 μm (left), 5 μm (right). **h** $\Delta F/F$ traces for the three neurons from **g**, comparing raw and FAST-denoised signals. Red box is magnified in (**i**). **i** Enlarged view from (**h**), with raw and FAST-denoised traces overlaid for direct comparison.

critical roles in brain function. However, the data processing for the volumetric Ca$^{2+}$ signaling is complicated and time-consuming, which limits the implementation of this imaging paradigm in the broader field. To address this challenge, we used FAST to enhance the quality of volumetric imaging data of cortical astrocytes in the somatosensory cortex (layer 2/3). Astrocytes were labeled with adeno-associated virus (AAV, rAAV-GfaABC1D-GCaMP6s-P2A-tdTomato-WPREs): tdTomato was used to label cellular structures, including somas, branches, and processes, while GCaMP6s was used to visualize intracellular Ca$^{2+}$ signals. As shown in Fig. 4a, two-photon volumetric fluorescence imaging was conducted using a resonant scanner for xy-plane scanning, combined with a fast piezo objective stage for z-axis scanning (see details in "Methods"). The imaging data were acquired in the dimensions of $2 \times (512 \times 512 \times 25) \times 336$ (channel-(x-y-z)-t).

FAST was employed separately for the structural and functional imaging channels due to differences in fluorescence intensity. Specifically, frames from each imaging plane in the volumetric stack were extracted and split into individual time-lapse stacks (xy-t) for each z-plane. These stacks were used to train two independent FAST models, enabling effective denoising of both structural and functional imaging data. For functional imaging, we compared the Ca$^{2+}$ fluorescence channel before and after FAST denoising (Fig. 4b, Supplementary Section 12 and Supplementary Video 5). Representative frames from the $z = 15$ slice demonstrated that FAST enhanced the visualization of calcium activity events, which were overlaid in different colors to distinguish individual events. In the raw data, noise obscured many events and diminished the clarity of their spatial and temporal features. After FAST denoising, the number of detectable events increased, and their spatial boundaries became more defined, allowing for clearer identification and tracking of calcium dynamics over time. The denoised structural images revealed significantly improved clarity in cellular structures, including somas, branches, and fine processes, as well as a reduction in background noise (Supplementary Section 11 and Supplementary Video 5). This enhancement facilitated the identification of astrocytic morphology and peripheral glial structures, which were difficult to discern in the raw data. Subsequently, we conducted a more in-depth analysis based on functional channel imaging.

To quantitatively assess the impact of FAST on calcium event detection, we extracted event features from the plane at $z = 15$ using AQuA software[46] (Fig. 4c–e). The comparison revealed significant improvements in event area, perimeter, and circularity after FAST denoising. Specifically, the denoised data exhibited an 8.15-fold increase in event area (from $2.41 \pm 1.05$ μm$^2$ to $19.65 \pm 10.23$ μm$^2$) and a 2.07-fold expansion in perimeter ($4.55 \pm 0.79$ μm vs $9.41 \pm 2.63$ μm), demonstrating enhanced boundary delineation capabilities. Concurrently, circularity measurements showed a 17.4% improvement ($1.84 \pm 0.08$ vs $2.16 \pm 0.13$), aligning more closely with expected calcium event morphologies. Statistical analysis using the Kolmogorov-Smirnov test confirmed these parametric differences ($P < 0.001$ for all

three features), with results presented as mean ± SEM. These findings demonstrate that FAST substantially improves the detection and quantification of calcium events, enabling more precise analysis of their spatial and temporal properties.

To further investigate the spatiotemporal distribution of calcium events, we analyzed volumetric imaging data of a single astrocyte (see Fig. 4f and details in "Methods"). Each calcium event was represented as a circle, with its spatial center corresponding to the event's centroid and its diameter scaled according to the event's volume. The duration of each event was color-coded to reflect temporal dynamics. In the raw data, the distribution of events appeared sparse and lacked clear spatial organization due to noise. In contrast, FAST-denoised data revealed a denser distribution of calcium events.

## Discussion

Noise presents a significant challenge in intravital neural observation, with inherent shot noise setting the ceiling for the SNR in fluorescence imaging, thereby constraining resolution, speed, and sensitivity. Effective denoising is crucial, and different imaging scenarios require tailored approaches: structural imaging benefits from high spatial resolution, while functional imaging demands high temporal resolution. Self-supervised deep learning methods have emerged as powerful tools for denoising by leveraging spatiotemporal redundancy in raw data. However, existing approaches often involve complex training and resource-intensive processes, and it remains challenging to provide the fast, high-quality denoising techniques needed for neuroscience research to support real-time data analysis and closed-loop neural modulation. Unlike previous approaches that primarily focus on increasing network complexity to extract spatiotemporal correlations, FAST introduces a paradigm shift in information utilization by employing a frame-multiplexed spatiotemporal sampling strategy. This shift enables the use of a much simpler 2D network architecture, fundamentally distinguishing FAST from prior methods. We introduce FAST, a universal real-time self-supervised denoising pipeline for diverse neural imaging scenarios. FAST uses a lightweight 2D convolutional network to balance temporal and spatial redundancies, achieving rapid denoising speeds beyond 1000 FPS, and extreme to 2100 FPS. It has been validated across various imaging scenarios and subjects, demonstrating superior denoising performance and real-time capability. FAST enhances cellular morphology restoration, neuron segmentation, and 3D calcium event extraction while preserving spike shapes and improving voltage transient correlations with electrophysiological recordings. A user-friendly GUI further facilitates training, offline inference, and real-time online inference, enabling state-of-the-art denoising of user data.

FAST, with its spatiotemporal balanced training strategy, achieves denoising speeds exceeding 1000 FPS on a single GPU using only a lightweight 2D convolutional network. Previous denoising methods for fluorescence time-lapse imaging often relied on large-scale

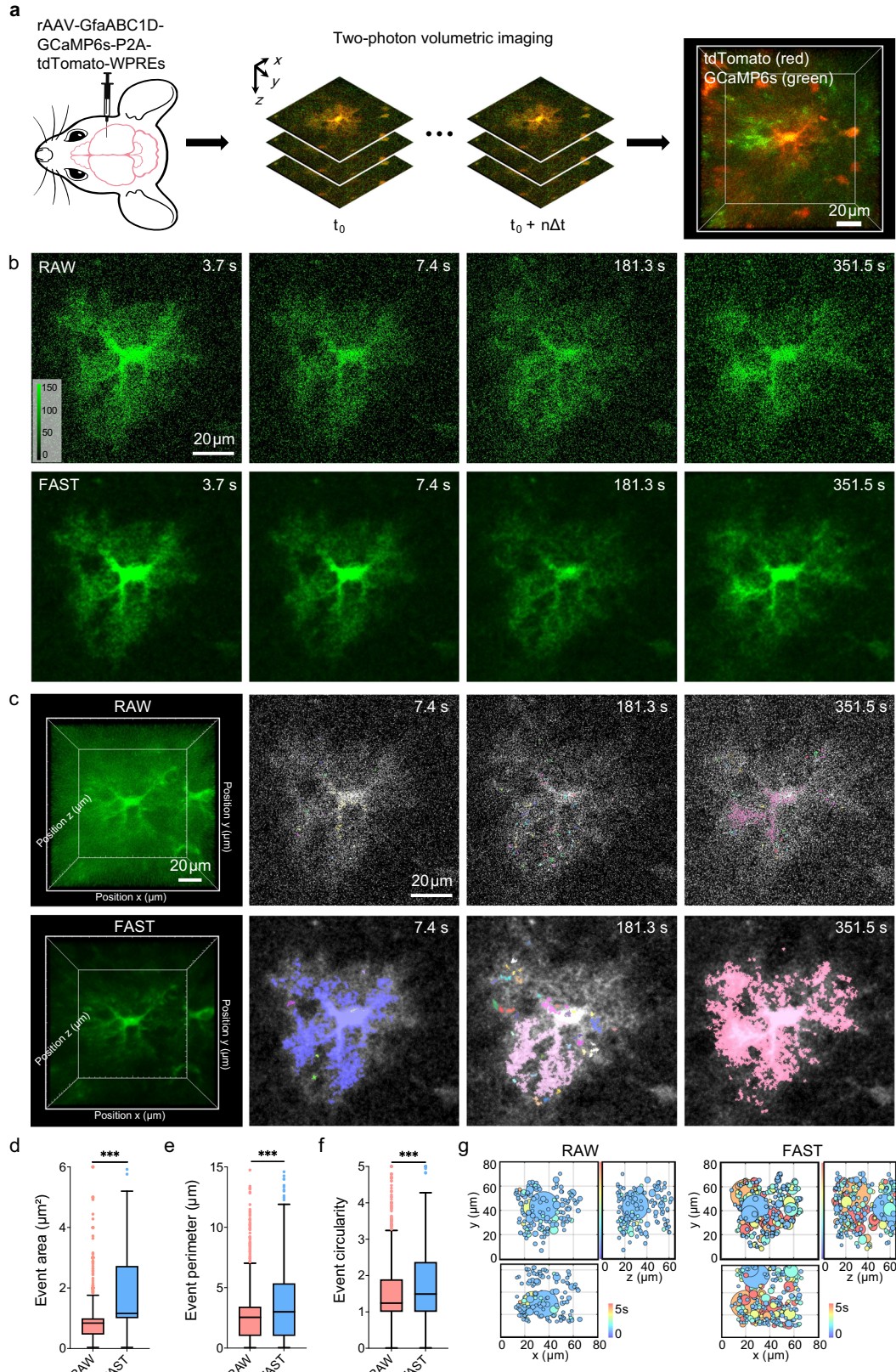

network structures, such as 3D convolutional networks, Transformers, or combinations of multiple architectures[15,16,21,24]. While these complex models have strong representational capacity, our results suggest that, within the scope of this study, such complexity does not bring significant advantages. It is important to note that the feasibility of using an ultra-lightweight 2D CNN is not achieved by incremental

architectural tweaks, but rather is made possible by our unique FAST sampling strategy, which intelligently balances spatial and temporal information during training. Our ablation studies show that increasing model complexity leads to only marginal improvements in denoising performance, while substantially increasing computational and memory costs. The flexibility of FAST is rooted in two key tunable

**Fig. 4 | FAST improves astrocytic Ca²⁺ events quantification in real-time spatiotemporal volumetric imaging. a** Two-photon imaging of cortical astrocytes was performed in awake mice. Cortical astrocytes were virus-labeled with GCaMP6s (denoting Ca²⁺ signals, in *green*) and tdTomato (denoting morphology, in *red*) fluorescence channels. **b** Direct comparison of raw and FAST-denoised images at representative time points. Images from the z = 15 plane are displayed for four selected frames (3.7 s, 7.4 s, 181.3 s, and 351.5 s), with the top row showing the raw data and the bottom row showing the corresponding FAST-denoised results. All images are displayed using an identical intensity scale, and a unified color bar indicating signal intensity is provided in the lower left corner of the first panel. This side-by-side comparison illustrates that FAST denoising effectively suppresses noise and enhances the visibility of astrocytic Ca²⁺ signals, particularly in regions with low signal-to-noise ratio. Scale bar, 20 μm. **c** Comparison of the raw data and data processed with FAST. The first row shows the raw data, and the second row shows the FAST-denoised data. The first column on the left displays the volumetric view of the calcium imaging channel. The second to fourth columns show images of the z = 15 slice at three representative frames (7.4 s, 181.3 s and 351.5 s), with overlaid colors indicating calcium activity events in the current frame, extracted using the AQuA method[46]. Different events are distinguished by different colors. In the raw data, extracted events were fragmented (a single continuous event erroneously detected as multiple separate segments), incomplete (detected events failing to capture the full spatiotemporal extent of the underlying signal), and often missing (true events not detected at all). After FAST denoising, the extracted events appeared more continuous, complete, and accurately defined, exhibiting spatiotemporal characteristics that align more closely with human perception. Scale bar, 20 μm. Quantification of event features extracted from the z = 15 plane in (**b**) for RAW and FAST data, analyzing all events across n = 336 frames. The box-and-whisker plots show the comparison of event area (**d**), event perimeter (**e**), and event circularity (**f**). Statistical significance was assessed using the two-sided Kolmogorov-Smirnov test (***P < 0.001). **g** The spatiotemporal distribution of Ca²⁺ events within the astrocytic territory. It displays the spatiotemporal distribution map of Ca²⁺ events extracted based on raw data (*left*) and FAST-processed data (*right*). Each circle represents a Ca²⁺ event, with the center coordinates corresponding to the event's spatial center and the diameter (μm) scaled according to the event's volume. The color scale denotes the duration of individual Ca²⁺ events (ranging from 0 to 5 s).

parameters: the temporal window width (*C*) and the temporal shift step (*S*). These parameters provide users with precise control over the method's performance based on their specific data and application needs. The parameter *S* primarily governs the trade-off between processing speed and denoising fidelity. As our analysis shows (Fig. 1b), increasing S dramatically boosts processing speed, though this comes at the cost of a moderate decrease in performance metrics like SSIM. This offers a clear choice: a larger *S* can be employed for real-time applications where throughput is critical, while a smaller *S* should be selected for offline analyses prioritizing maximum image fidelity. The parameter *C* controls the temporal context provided to the model. For datasets with very fast signal dynamics, we recommend using a smaller *C*. This narrows the temporal receptive field, preventing distinct, rapid events from being overly smoothed and thus preserving crucial temporal precision. Therefore, by jointly tuning *S* and *C*, users can flexibly adapt FAST to a wide range of imaging scenarios, balancing computational demands with the need to faithfully represent complex signal dynamics. The performance of FAST meets the real-time denoising demands of most imaging systems, while still leaving time windows available for additional post-processing operations, such as cluster analysis of large-scale neuronal imaging, real-time spike inference, functional connectivity analysis, online activity pattern recognition, optogenetic stimulation feedback, and automated ROI selection and tracking[42–46]. We note that for other imaging modalities or tasks characterized by more complex noise patterns, higher model complexity or specialized architectural designs may be required to achieve optimal denoising performance. To further clarify the boundaries of FAST, we systematically evaluated its robustness to motion artifacts and broadband noise. As detailed in Supplementary Section 13 and Supplementary Fig. 11, FAST maintains high denoising performance under no or slow motion, but its effectiveness declines with vigorous motion. Therefore, in practical applications involving substantial motion, we recommend combining FAST with image registration or motion correction as a preprocessing step.

In selecting our network architecture, we also carefully considered a range of modern efficient architectures, including recent lightweight Transformer models[47,48]. Currently, CNN-based solutions still offer clear advantages in inference speed, computational efficiency, and implementation simplicity for the extreme real-time and low-latency demands of high-frame-rate applications[49–51]. We therefore adopted a lightweight CNN as the backbone of FAST. Nevertheless, we fully acknowledge the distinct advantages of many of those advanced efficient models, particularly lightweight Transformers, with their powerful ability to model long-range dependencies and capture global contextual information. This capability is especially promising for tackling complex, non-local noise patterns. As these architectures are continuously optimized for greater computational efficiency, they are poised to become a compelling alternative for real-time denoising.

Existing self-supervised time-lapse denoising methods rely on temporal redundancy to recover true signals from noisy data. These methods are based on the assumption that information carriers maintain ergodicity across adjacent temporal coordinates. This foundational assumption, however, confronts inherent limitations when the temporal sampling rate is low or the signal's dynamics are fast. Hence, in practical ultra-high speed imaging acquisition, such as neuronal voltage imaging, the dynamics between a spike window are too sharp to maintain the assumption after temporal subsampling, and further lead deviations that can lead to distortion and smoothed spatial structure of recovered signals. Most current methods seek to maximize the temporal input range, aiming to enhance denoising performance by capturing potential signal features. However, a wider temporal input range often contains several dynamics. A self-supervised approach relying solely on temporal redundancy would cause a phantasm of temporal causality, wherein signal restoration highly depends on future signals, especially in 3D convolutional networks. This temporal causality phantasm inevitably introduces uncertainties in ultra-high speed imaging, compromising the authenticity of the restored signal. This conceptual advance allows FAST to flexibly adapt to diverse signal dynamics by decoupling the reliance on strict temporal continuity, thus overcoming the limitations of traditional self-supervised frameworks in handling rapid or non-ergodic neural events. FAST mitigates this issue by incorporating adjustable trade-off parameters that constrain the temporal receptive field and expand the spatial receptive field. By tuning the information-receiving range, FAST effectively minimizes interference at its source, preserving the authenticity of neural signals while reducing noise. This controlled approach enables the rapid and faithful restoration of high-fidelity signals without relying on extensive spatiotemporal inputs.

Our well-designed frame-multiplexed spatiotemporal learning strategy equips FAST with a universal self-supervised denoising capability, demonstrated to be effective across a wide range of biological samples, including zebrafish and mice. It is versatile in its application, supporting multiple imaging modalities, such as calcium imaging, voltage imaging, and volumetric imaging, as well as various imaging techniques like confocal microscopy, light-sheet microscopy, and multi-photon microscopy. Beyond this, we believe FAST has the potential to be implemented in other biomedical optical imaging techniques, offering improvements in depth, resolution, and speed. Therefore, we envision FAST as providing universal ultra-fast denoising integrated with imaging systems and experimental requirements, thereby opening up new possibilities for advanced and verifiable biomedical research.

## Methods

### Spatiotemporal sampling strategy

In FAST, we employ a spatiotemporal redundancy sampling strategy to generate training pairs, as detailed in Supplementary Fig. 3. For a training stack with dimensions $X \times Y \times t$ pixels (where $X$, $Y$, and $t$ represent the height, width, and length of the input image stack, respectively), we first perform temporal sampling. This involves dividing the stack into input stack $X_i$ and target stack $Y_i$ using a sliding window approach. Both the input and target stacks have dimensions H × W × C, where C and S denote the window width and stride of the sliding window, respectively, with $S \leq C$. Next, we apply spatial sampling to $X_i$ and $Y_i$ for each image, using a method similar to sparse convolution computation in neural networks. This results in two sub-stacks $G_1(X_i)$, $G_2(X_i)$ from $X_i$ and $G_1(Y_i)$, $G_2(Y_i)$ from $Y_i$. The sparse convolution uses a $2 \times 2$ pixel window with a stride of 2, and the convolution kernel is randomly selected from a list of 16 mask cells. Each mask cell is a $2 \times 2$ binary matrix with two elements set to 1, facilitating the selection of adjacent pixel positions. We employ the same random seed for the identical windows in $X_i$ and $Y_i$, ensuring that the original pixel positions of $G_1(X_i)$ and $G_1(Y_i)$ match, and those of $G_2(X_i)$ and $G_2(Y_i)$ also match. Additionally, $G_1(X_i)$ and $G_2(Y_i)$ conform to spatiotemporal adjacency, as do $G_2(X_i)$ and $G_1(Y_i)$. Finally, we select $X_i$ and $G_1(X_i)$ as the input stack, and $Y_i$ and $G_2(Y_i)$ as the target stack, for the self-supervised training of the denoising network.

### Network architecture and loss function

**Network structure.** The FAST network is a meticulously designed ultra-lightweight 2D U-Net, as illustrated in Supplementary Fig. 3. Similar to the original U-Net[52], our network architecture employs an encoder-decoder structure with skip connections. In our network, we use two encoding blocks to reduce the dimensionality. Each encoding block consists of a $3 \times 3$ group convolutional layer (with each group processing 2 channels) followed by a BatchNorm layer, a ReLU activation, and a $3 \times 3$ max-pooling layer. To restore the input dimensions, we use two decoding blocks, each containing a nearest-neighbor interpolation, followed by a $3 \times 3$ group convolutional layer (with each group processing 2 channels), a BatchNorm layer, and a ReLU activation. Group convolutions reduce the number of parameters and computational cost while enabling independent feature extraction within each group, which helps mitigate interference between groups and avoids artifacts caused by temporal signal aliasing. At each resolution depth, the number of channels in each feature map is 64. We employ skip connections to connect the output of the second encoding block to the input of the corresponding decoding block.

Compared to the original U-Net, we have made two principal modifications. First, we reduced the number of downsampling steps to two. Downsampling increases the receptive field, allowing the convolutional kernels to extract features over a larger image area. However, as the number of downsampling steps increases, so does the number of network parameters. Our spatial subsampler can be regarded as a single downsampling step that expands the receptive field, thereby allowing us to reduce the number of downsampling steps in the U-Net to shrink the network. Second, we eliminated the top-level skip connection. Skip connections can fuse low-level and high-level features, but in denoising tasks, high-level features are often noise that needs to be suppressed. Therefore, removing the top-level skip connection can improve denoising performance. By minimizing the network depth, the number of channels per feature map, and the number of skip connections, our network trains with only approximately 0.013 M parameters.

**Loss function.** We optimize the parameters of FAST using a linear combination of two loss functions, corresponding to scale self-constraining and spatiotemporal self-supervising. Let $X_i$ denote the input stack, $Y_i$ the output stack, and $f(\cdot)$ the network, $G_1(\cdot)$, $G_2(\cdot)$ the

sampling strategies described in the sampling method. We enforce spatiotemporal adjacency and scale consistency constraints. Scale consistency ensures the model maintains performance between input and sampled output, defined as $L_{SC} = \|f(G_1(X_i)) - G_1(f(X_i))\|_2^2$. Spatiotemporal adjacency utilizes information from adjacent pixels in space and time to achieve denoising, defined as $L_{ST} = \|f(G_1(X_i)) - G_2(Y_i)\|_2^2 + \|P(f(G_1(X_i))) - P(G_2(Y_i))\|_1$. Here, $P$ calculates the average intensity over randomly sampled $20 \times 20$ patches across 10 regions, used to maintain the consistency of intensity trends over time in multi-frame images. The total loss is formulated as $L_{total} = L_{SC} + L_{ST}$. To ensure stable training, gradients are not propagated through $f(X_i)$.

**Training and inference.** The network was trained on an NVIDIA RTX A6000 GPU with CUDA 12.2. The entire input image stack ($X \times Y \times t$) is treated as a single sample, which is sequentially fed into the network after being split along the temporal dimension $t$ according to the spatiotemporal sampling strategy. Data augmentation methods such as random flipping and 90° integer multiple rotations are employed. By default, the network is trained for 100 epochs, where each epoch allows the network to process the full inflow of an image. The batch size is set to 1, and the optimizer used is ADAM with a learning rate of $1 \times 10^{-4}$ and a weight decay of $1 \times 10^{-4}$, utilizing AMSGrad. To ensure reproducibility, all relevant libraries, including NumPy and PyTorch, have their random seeds initialized to 123. For the denoising of 3D volume imaging ($X \times Y \times Z \times t$), each image stack of the imaging planes is treated as $Z$ samples and collectively serves as the input for network training.

### System latency and pipeline integration

Achieving true real-time performance requires considering the entire pipeline, including data acquisition, transfer, motion correction, and other downstream steps, whose latencies can be substantial and hardware-dependent. A key advantage of FAST is that its processing time represents only a small fraction of this budget (e.g., <1–2 ms for a $512 \times 512$ frame versus ~33 ms for a 30 Hz acquisition), leaving ample headroom for more demanding operations and ensuring it does not become a bottleneck in diverse real-time systems.

### Neuron segmentation and evaluation metrics

**Segmentation.** Cellpose is a versatile deep learning model designed for cell segmentation, trained on a diverse dataset, including two-photon calcium imaging data from mice. During its training process, Cellpose uses average projection on calcium imaging data to create single-frame images for segmentation. However, for the Neurofinder dataset, using average projection would result in the loss of many neurons. To address this, we applied maximum intensity projection to the fluorescence time-lapse images, aiming to retain as many neurons as possible before feeding them into the Cellpose model. Several configuration options were set prior to input: we selected cytoplasm segmentation, used the gray channel, opted not to use a nuclear channel, and set the average cell diameter to 20 pixels.

For segmentation, we selected Cellpose as a unified model in this study. Cellpose is a general-purpose segmentation tool whose training data includes two-photon calcium imaging neuronal segmentation, making it applicable to our task. Its use of a single adjustable parameter (target cell diameter) also facilitates standardized evaluation across datasets. We note that other supervised segmentation methods may achieve good performance but typically require retraining or fine-tuning on specific datasets, which is not aligned with our goal of an unbiased and consistent evaluation pipeline.

**Evaluation metrics.** We compared the segmentation results of the raw images and the denoised images with manual ground truth labels to evaluate the improvement in segmentation brought by the denoising

methods. The segmentation results were quantitatively assessed using three metrics: Accuracy, Recall, and F1 score, defined as follows:

$$Accuracy = \frac{TP + TN}{TP + TN + FP + FN} \tag{1}$$

$$Recall = \frac{TP}{TP + FN} \tag{2}$$

$$F_1 = \frac{2 \times Precision \times Recall}{Precision + Recall} \tag{3}$$

where $TP$ is true positives, $TN$ is true negatives, $FP$ is false positives, and $FN$ is false negatives.

## System setup for two-photon imaging

The imaging system for two-photon imaging is built upon a commercial multi-photon microscope (DeepVision-2P, MicroLux, China), which employed a 920-nm femto-second pulsed laser (ALCOR-920, 100-fs pulse duration, 80-MHz repetition rate, Sparks Lasers) as the light source. A 5× beam expander (GBE05-B, Thorlabs) was subsequently used to increase the beam diameter. The collimated and expanded laser beam was directed to an 8-kHz resonant scanner (CRS-8K, Cambridge Technology). A 1:1.25 relay system was configured to establish a conjugate relationship between the resonant scanner and a linear scanner (6215 K, Cambridge Technology). A scan lens ($f$ = 80 mm) and a tube lens ($f$ = 200 mm), were used to expand the light beam to fill the back aperture of a 16× water immersion objective lens (N16XLWD-PF, Nikon) after going through a long-pass dichroic mirrors (T750lpxrxt-UF3, Chroma).

The emitted fluorescence traversed the long-pass dichroic mirror, a short-pass filter (ET720sp-2p8, Chroma), a collective lens ($f$ = 100 mm), and was subsequently separated by a secondary dichroic (T556lpxr-UF3, 42 × 60 × 3 mm, Chroma) into green and red color channels. Red and green fluorescence were then focused by lenses with focal length of 50 mm and directed onto high-sensitive GaAsP photomultiplier tubes (PMTs, H10770PA-40, Hamamatsu), respectively. The PMT signals were amplified with a 60-MHz preamplifier (TIA60, Thorlabs) and digitized using a data acquisition card (vDAQ, MBF Bioscience). Scanning and data acquisition processes were controlled using ScanImage 2022b (MBF Bioscience).

## Animal experiments

**Animal preparation.** Wild-type male C57BL/6 mice (2–3 months old, weighing 25–30 g) were obtained from the Institutional Laboratory Animal Center (Shanghai, China). The mice were housed in groups on a 12:12 light/dark cycle with free access to food and water. All procedures involving animals were conducted in accordance with institutional guidelines for animal welfare and approved by the Animal Care and Use Committee, Fudan University.

**Surgery preparation.** Mice were anesthetized with an intraperitoneal injection of sodium pentobarbital (50 mg/kg body weight). After retracting the scalp, we carefully removed the fascia from the upper cranial surface using a razor blade. A metal plate was affixed to the skull, exposing the right hemisphere. A 4 mm diameter craniotomy was drilled into the somatosensory cortex, leaving the dura mater intact. The rAAV-GfaABC1D-GCaMP6s-P2A-tdTomato-WPREs virus (BrainVTA, China; 300 nL of 5 × 10^12 VG/mL) was then injected into the targeted region (AP: −1.5 mm; ML: −2 mm; DV: −0.3 mm) to label astrocyte structure with red fluorescence and Ca²⁺ signals with green fluorescence, respectively. The craniotomy was covered with a glass coverslip, which was secured to the skull using super glue (Ergo5800, Kisling, Switzerland). Mice were allowed to recover for at least 2 weeks before in vivo imaging. To minimize the movement artifacts during awake imaging, the animals were trained daily for adaptation for the imaging setups during 5–7 days prior to the experiments. The first training session lasts 15 min, gradually prolonging into 1 h in the following days.

## Two-photon volumetric imaging of cortical astrocytic morphology and Ca²⁺ signaling

**Imaging protocols.** Imaging was performed using a two-photon microscope with resonant scanning at 30 Hz, where the objective lens was axially scanned to capture 3D volumes. The imaging volume was scanned at a resolution of 512 × 512 pixels across 25 layers with a 2.4 µm increment in the z-axis, achieving a lateral voxel size of 0.2 µm, an axial voxel size of 2.4 µm and an effective volume rate of 1.2 Hz.

**3D-view display.** We used Imaris 9.0 (Oxford Instruments) to visualize all volumetric imaging data of astrocytes. The original low SNR data and denoised data were imported into Imaris for pseudo-color rendering, where the red channel imaged astrocytic morphology, the green channel imaged intracellular Ca²⁺ signaling. The reconstruction mode is selected for maximum intensity projection.

## Astrocyte Ca²⁺ event extraction

**Event extraction for a single plane.** Astrocyte calcium events were extracted from individual planes using the AQuA software analysis package[46]. Masks were created in AQuA to exclude regions outside the cells, the minimum size of the connected region was set to 4, the intensity threshold scaling factor was set to 7, and the standard deviation of the Gaussian filter was set to 0.2. To enhance image detail, the event area masks were overlaid onto the images in Matlab. In this visualization, colors are randomly assigned to events on a per-frame basis; thus, within a single frame, the same color represents the same event, but the same color in different frames does not indicate correspondence between events across time.

**Event extraction on volume.** We implemented the extraction of volumetric events using Matlab by first calculating the $\Delta F/F$ of the Ca²⁺ fluorescence channel data. Next, we performed k-means clustering on the data of each frame to determine the fluorescence signal threshold. Events were detected using connected component analysis, and both the volume and intensity of each event were calculated. The event data and global frame information for each frame were saved. The event data across all frames were then merged, and the intersection over union (IoU) ratio was calculated. Events were merged based on the IOU, and the merged event information was stored. Finally, the duration of each event was extracted from the merged data, and key information, such as spatial location, volume, and intensity of the events, was calculated.

## GUI of FAST for real-time image acquisition and denoising

The GUI integrates MATLAB and Python to deliver high-speed image acquisition and real-time denoising using FAST. The architecture comprises two core components: a MATLAB-based application, which uses ScanImage for capturing images, and a Python-based module designed for real-time image denoising through deep learning. The system operates by first acquiring images via the MATLAB application, which then buffers the data and transmits it to the Python module using HTTP/HTTPS protocols. The Python module processes these images to reduce noise and subsequently saves the denoised results to hard disk.

The MATLAB interface allows users to configure acquisition parameters and control the image capture process, providing real-time updates and status information. On the other hand, the Python interface offers a dynamic visualization of the processed images, complete with interactive tools for adjusting display settings and monitoring processing status. The implementation leverages MATLAB for

acquisition and Python with frameworks PyTorch for denoising. Key algorithms include neural network architectures optimized for balancing denoising with image detail preservation. Data transfer between components takes place via secure and efficient HTTP/HTTPS communication, the format of requests and responses follows the JSON standard, while testing includes unit, integration, and performance evaluation to ensure functionality and real-time processing capabilities. Challenges such as data transfer latency were addressed through optimized buffering techniques and improved protocol settings, ensuring the smooth and effective operation of the FAST system.

## Statistics and reproducibility

The sample size and statistical details are reported in the figure legends, figure panels, and main text for each experiment. All box-and-whisker plots are presented in the standard Tukey format. The boxes represent the interquartile range (IQR) between the first and third quartiles, with the line inside the box indicating the median. The lower whisker extends to the smallest data point greater than or equal to the first quartile minus 1.5 times the IQR, and the upper whisker extends to the largest data point less than or equal to the third quartile plus 1.5 times the IQR. Outliers are indicated by dots.

## Reporting summary

Further information on research design is available in the Nature Portfolio Reporting Summary linked to this article.

## Data availability

All raw datasets and relevant materials used in this study are publicly available on Zenodo at https://doi.org/10.5281/zenodo.15872025.

## Code availability

The underlying code for this study is available at https://github.com/FDU-donglab/FAST.

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

## Acknowledgements

We thank Y. Hu for contributing to imaging data collection. We also thank the Advanced Biomedical Imaging Facility in Hubei for the support. This work was supported in part by the National Key R&D Program of China (2022YFF0708700 for B.D.), National Natural Science Foundation of China (82371488, 82422028 for F.D., 32200919 for C.K., 32271023 for W.F., 62375095, 62375096, and 32361133552 for D.L. and D.Z.), Shanghai Pilot Program for Basic Research (22TQ020 for B.D.), Natural Science Foundation of Shanghai (22ZR1404300 for B.D.), Shanghai Science and Technology Innovation Action Plan (22S31905500 for B.D.).

## Author contributions

Y.W. and Y.G. contributed to the conceptualization, coding, comparisons, and visualization. J.W., A.X., C.K., and D.L. built experimental setups and performed data collection. W.F., D.Z., F.D., and B.D. supervised the project, contributed to the conceptualization, and designed experiments. Y.W., Y.G., F.D. and B.D. wrote the manuscript. All authors have read and approved the manuscript.

## Competing interests

B.D. is a founder and equity holder of MicroLux (Shanghai) Intelligent Science & Technology Co., Ltd. and Lishi Intelligent Science & Technology (Shanghai) Co., Ltd. B.D., Y.W., and Y.G. submitted patent applications related to the FAST technology described in this work. All other authors declare no competing interests.
