## [Transparent Peer Review file · Nature Communications]

Real-time self-supervised denoising for high-speed fluorescence neural imaging

Corresponding Author: Professor Biqin Dong

Version 0:

Reviewer comments:

Reviewer #1

(Remarks to the Author)

This manuscript introduces FAST (FrAme-multiplexed SpatioTemporal learning strategy), a self-supervised deep learning framework for real-time denoising of high-speed fluorescence neural imaging data. The authors claim FAST utilizes an ultra-lightweight 2D convolutional neural network (CNN) with only 0.13 million parameters, enabling processing speeds exceeding 1,000 frames per second (FPS), and up to 2,100 FPS. The method is validated on in vivo calcium, voltage, and volumetric time-lapse imaging data from mice and zebrafish. Key contributions include the spatiotemporal sampling strategy, the lightweight architecture, remarkable processing speed, and a graphical user interface (GUI) for integration into imaging workflows. The authors demonstrate improved neuron segmentation, preservation of rapid voltage transients, and enhanced 3D calcium event extraction.

Strengths:

- 1) The reported processing speeds are indeed impressive, particularly for applications like closed-loop experiments.
- 2) The use of a 0.13M parameter model is noteworthy for achieving high performance, minimizing computational overhead and memory requirements.
- 3) The validation across three imaging modalities and different model organisms demonstrates the potential versatility of FAST.
- 4) The self-supervised nature, leveraging spatiotemporal redundancies and avoiding the need for clean ground truth data for training, is highly practical for many experimental setups.

Major Concerns and Points for Revision:

- 1) The authors attribute the success of their 0.13M parameter model to the "meticulously designed ultra-lightweight 2D U-Net", specific architectural choices, and the "spatiotemporal balanced training strategy". They also state that their "spatial subsampler can be regarded as a single downsampling step that expands the receptive field, thereby allowing us to reduce the number of downsampling steps in the U-Net to shrink the network."

While these explanations are provided, the link between these specific choices and the exceptional performance on diverse, complex biological data could be strengthened. An ablation study demonstrating the impact of each key architectural modification (e.g., with/without group convolutions, with/without the top-level skip connection, varying downsampling depths) on both performance and speed would provide more convincing evidence for why this specific lightweight design works so well.

The Discussion mentions that "complexity is unnecessary." While this is a strong claim, a more nuanced discussion comparing the trade-offs between model complexity, receptive field, and the nature of noise/signal in fluorescence imaging would be beneficial.

2)The manuscript compares FAST with DeepCAD-RT, SRDTrans, DeepVid, and SUPPORT. For the calcium imaging dataset, it's stated that raw videos were "denoised using five different methods... followed by temporal maximum intensity projection (MIP) and segmentation using Cellpose."

It is crucial to detail how these competitor models were run. Were their pre-trained models used (if available and applicable)? Or were they retrained on the specific datasets presented in this manuscript? If retrained, how were their hyperparameters selected and optimized? The concern that FAST might be meticulously tuned for the presented datasets while competitor models are not (or run with default/suboptimal parameters) is valid and needs to be explicitly addressed to ensure fair comparison. This is particularly important for self-supervised methods where training is dataset-specific.

The authors should state whether they used the official implementations of the compared methods and what, if any, parameter tuning was performed for them on the datasets used in this study.

3)The GCaMP6s data used for evaluation is from Ref. 32, which links to the Neurofinder challenge (neurofinder.codeneuro.org). Neurofinder is a recognized public benchmark. The authors perform their own segmentation and evaluation against manual ground truth. It would strengthen the claims if they could (a) compare their quantitative segmentation results to published/leaderboard results for Neurofinder for the same dataset instance if available, or (b) clearly justify why their specific evaluation protocol is a robust alternative for comparison. It is more recommended to benchmark on a widely adopted, independent public challenge dataset with a leaderboard.

4) The manuscript states: "Data are available from the corresponding author upon reasonable request." While code is available on GitHub, this data availability statement is a significant barrier to independent reproduction and verification, which is critical for a journal like Nature Communications.

Recommendation: The authors should be strongly encouraged to deposit all raw data used to generate the figures into a publicly accessible repository (e.g., Zenodo, Figshare, a specialized neuroscience data archive) without requiring a request. This would substantially bolster the claims of reproducibility.

5) The authors acknowledge that previous methods have used Transformers. Their argument for not using a more complex model like a Transformer is primarily centered on the desire for an "ultra-lightweight" network to achieve extreme processing speeds, stating in the Discussion, "we argue that such complexity is unnecessary."

This is a reasonable justification given their primary goal of real-time processing. However, a brief discussion acknowledging if any lightweight Transformer variants were considered and why the CNN approach was ultimately favored might be helpful.

Overall, the manuscript presents FAST, a compelling method for real-time denoising of fluorescence neural imaging data with impressive speed and a lightweight architecture. The work is generally well-presented, and the demonstrations across diverse imaging modalities are valuable.

However, for publication in a high-impact journal like Nature Communications, the concerns regarding the justification of the small model's efficacy (potentially via ablation studies), the fairness of comparisons to other methods (hyperparameter tuning details for competitors), and especially data availability for reproducibility need to be thoroughly addressed.

(Remarks on code availability)

The FDU-donglab/FAST repository implements a self-supervised denoising algorithm for 3D image time-series, using a 2D U-Net with a "frame-multiplexing" strategy and a composite loss function. The core FAST algorithm appears to be implemented as described.

Reproducibility of published results is partially supported by aspects like configuration saving and random seed management. However, the lack of specified library versions, exact dataset details, and some hardcoded parameters might require effort to precisely match paper outcomes.

As a community resource, the code's modular design is a plus. However, it does not have a comprehensive test suite. The lack of comprehensive tests also impacts its robustness and maintainability, which is often a requirement for good software. Addressing these documentation and safety issues would greatly enhance its value to the community.

Reviewer #2

(Remarks to the Author)

The manuscript entitled "Real-time self-supervised denoising for high-speed fluorescence neural imaging" by Wang et al. described a real-time image denoising technique for functional neuronal imaging. The authors demonstrated the method on multiple calcium and voltage imaging datasets. This is an interesting work, and I appreciate the primary novelty of the technique's ability to perform real-time denoising.

Major comments:

1. From the reported demonstrations, I fail to see the necessity of real-time denoising, as all results were obtained post hoc. It would strengthen the paper if the authors could include at least one experiment that directly benefits from real-time processing.
2. Related to point 1, all Results sections are titled “Real-time ... with FAST”. In my understanding, real-time implies that denoising is performed concurrently with image acquisition. However, the demonstrated experiments are not the case. The authors may consider describing their method as fast.
3. Comments on Fig. 2. In Fig. 2h, the blue circular overlays are visually distracting. For example, it is difficult to assess whether FAST performs better on Cells 2 and 3. For Cell 2, all methods seem to produce discontinuous morphology by eye. For Cell 3, some discontinuous circle-like structures are also visible with SRDTrans and DeepVid, but the blue circle on FAST makes it hard to visually compare the results. Therefore unfortunately, I am not convinced that “..., for Cell 2, only SRDTrans and FAST enabled accurate segmentation, with other methods either over-smoothing the boundaries or introducing artifacts. Notably, only FAST successfully segmented Cell 3, recovering subtle neuronal structures obscured by noise.” The authors may consider using arrowheads instead. The authors also claimed that “FAST-denoised images revealed synaptic structures ...”, can the authors use arrows to indicate the synaptic structures? From the current images, they are not clearly visible. For Fig. 2i, how many neurons were used? In addition, SUPPORT produced some horizontal fringes – we have tested SUPPORT on our calcium datasets and didn’t observe such artifacts. Could the authors comment on this?
4. Comments on Fig. 4. In Fig. 4b, I am not sure whether the improved image quality is primarily due to denoising. I tried background subtraction by (1) removing a constant-value background and (2) using a 100/200-pixel-diameter rolling ball, and obtained images that appeared similar to those from FAST. Although I didn’t do detailed analysis, the authors may want to further verify that the improvement is not due to basic background removal. The overlaid colors are also distracting. The authors should at least show side-by-side comparisons of raw and denoised images without overlays, and provide a color bar indicating activity strength. In the caption, some words need further clarification: “extracted events were fragmented, incomplete, and often missing. After ... more continuous, complete, and accurately defined”. It is also not clear what the effective volume rate is. From Methods, the frame rate is 30Hz, and the data were acquired with $2 \times 512 \times 512 \times 25 \times 336$: does that mean the volume rate is $30/25=1.2$ Hz? Is this sufficient to capture GCaMP6s-reported calcium activities? Finally, in Methods: “The imaging volume was ... achieving a lateral resolution of 0.2 μm and an axial resolution of 2.4 μm .” This is not accurate – resolution is determined by the NA of the objective lens, sampling, aberration, and other factors.

Minor comments:

1. Line 427: “structrual” should be “structural”.
2. In line 677: the sentence “there is no correspondence between the same color across different images” is unclear.
3. The dataset from Ref 35 was acquired using SIM. The authors may want to clarify that it refers to background-rejection SIM, not super-resolution SIM.

(Remarks on code availability)

Reviewer #3

(Remarks to the Author)

The manuscript introduces FAST, a deep-learning method for real-time denoising in fluorescence microscopy, optimized through a compact neural network and a paired-frame, self-supervised training approach. FAST achieves high processing speeds (>1000 FPS) and preserves rapid neural signals, representing a valuable technical improvement over existing techniques. While the work is practically significant for high-speed neural imaging, its incremental advancement relative to previously published methods such as DeepCAD-RT (Li et al., Nature Biotechnology, 2023) limits conceptual novelty. The analysis presented is sound, but the interpretation should explicitly acknowledge limitations, particularly regarding sensitivity to motion artifacts and blood flow. Critically, the manuscript primarily shows post-processed examples rather than explicitly demonstrating the method’s real-time capabilities, such as closed-loop applications or live imaging.

Major comments

1. Technical Novelty: The manuscript represents an incremental advancement over existing methods such as DeepCAD-RT, primarily through enhanced computational efficiency and processing speed. To strengthen the manuscript, authors should explicitly highlight unique conceptual differences compared to prior methodologies.
2. Real-time Processing Demonstration: Although the manuscript emphasizes FAST’s high processing speeds as its key contribution, it predominantly presents post-processed results. Explicit demonstrations or quantitative evaluations of real-time performance (e.g., latency measurements, closed-loop imaging scenarios) are essential to convincingly justify its need.
3. Limitations in handling motion and broadband noise: FAST fundamentally relies on minimal motion between frames. Substantial motion and broadband noise such as bloodflow would significantly decrease its efficacy. The manuscript should address and acknowledge this limitation and state the necessity of incorporating image registration or motion correction as fast as this processing technique.
4. Robustness and Generalizability: Authors should clarify the range of conditions (noise levels, indicator brightness, neuron

types) under which FAST reliably performs. A thorough exploration of robustness would significantly strengthen claims of applicability.

5. Parameter Sensitivity and Trade-offs: The manuscript should address the sensitivity of FAST's performance to key parameters such as frame interval selection and spatial-temporal weighting, outlining potential trade-offs.

6. Biological Interpretability: The manuscript provides limited insight into how improved technical image quality meaningfully enhances biological interpretability. Demonstrating explicit biological or analytical advantages would strengthen the practical relevance and impact.

Minor comment.

Voltage imaging is typically not analyzed per pixel basis, as a practical note, signals are extracted from ROI as pixel averages. Moreover, the randomly selected pixels should be taken from the cell membrane and not the cytoplasm due to the way the sensor labels neurons.

Recommendation: Major revision or consideration for submission to a specialized, methods-oriented journal unless novelty and broad applicability are significantly enhanced.

(Remarks on code availability)

Version 1:

Reviewer comments:

Reviewer #1

(Remarks to the Author)

The authors have thoroughly addressed my concerns raised in the previous round of review. The manuscript has been significantly strengthened through the inclusion of new supporting data, expanded analyses, and important clarifications throughout the text. The authors' commitment to transparency and reproducibility is commendable, and the revised manuscript is a much more compelling and robust piece of work. I would recommend it for publication.

(Remarks on code availability)

The code appears to be of high quality. The documentation is clear, and the setup is straightforward, though I didn't install and run the code.

Reviewer #2

(Remarks to the Author)

The authors performed additional experiments and provided clarifications to some of my earlier concerns. Below are my remaining comments:

A. Comments on the necessity of doing real-time:

The authors conducted real-time denoising with an in vivo mouse brain imaging experiment. It demonstrated the speed advantage, but it didn't show how real-time denoising benefits biological studies. I agree with what the authors listed; these are, of course, practical values of real-time denoising, but the direct benefit was not demonstrated.

B. Comments on Figure 2:

This part includes 155 neurons demonstrating the segmentation performance and 3 neurons showing cell morphology and segmentation accuracy.

Overall, the current data do not support the statement, "FAST demonstrated superior performance in restoring neuronal morphology and improving segmentation outcomes compared to other methods." For the statistical analysis in Fig. 2i – All methods are mostly comparable except for SUPPORT – differences are really subtle.

For the three selected cells, given the non-significant statistical differences shown in Fig 2i, the most I can conclude is that in some cases, FAST works better. In addition, it looks like all the images in Fig. 2h were individually normalized (from 0/min to max, the authors may want to confirm this), except for the image with DeepCAD-RT, which was obviously much dimmer. The authors should check how images were normalized for fair comparisons.

Also, the marked dendrites can also be visualized using DeepCAD-RT and SRDTrans.

Lastly, I am not convinced by the explanation of why SUPPORT produced fringes – in the original paper, the authors (Eom et al. 2023, Nature Methods) already demonstrated calcium imaging applications. The lower frame rate should not be the reason.

(Remarks on code availability)

Reviewer #3

(Remarks to the Author)

The authors have adequately addressed several points in the revised manuscript, and it is much improved. While the authors have done a good job of addressing the Reviewer comments, I would like to point out an issue:

The abstract mentions "... Utilizing an ultra-light convolutional neural network, FAST enables real-time processing at speeds exceeding 1,000 frames per second...", but the revision shows calcium imaging data denoising at 30 Hz. Unless real online denoising of fast functional data is shown, I suggest reconsidering the wording of the initial claim, since it might appear confusing to the reader.

Moreover, I would suggest adding a "latency budget" with reasonable values for achieving true live denoising. It can include data readout, motion correction, registration, and the denoising algorithm. I believe having a reasonable expected number would be useful for people using all these methods.

(Remarks on code availability)

Point-by-point response to the reviewers' comments

Title: Real-time self-supervised denoising for high-speed fluorescence neural imaging

Author: *Yiqun Wang, Yuanjie Gu, Jianping Wang, Ang Xuan, Cihang Kong, Wei-Qun Fang, Dongyu Li, Dan Zhu, Fengfei Ding, and Biqin Dong*

We greatly appreciate the reviewers' positive evaluation of our work and their constructive comments and suggestions. We have carefully revised the manuscript to address all the concerns and improve the clarify of the manuscript. We highlight the revised sections with the red color in the revised manuscript. Please see our point-by-point response to reviewers' comments below:

Reviewer #1:

(Remarks to the Author)

This manuscript introduces FAST (FrAme-multiplexed SpatioTemporal learning strategy), a self-supervised deep learning framework for real-time denoising of high-speed fluorescence neural imaging data. The authors claim FAST utilizes an ultra-lightweight 2D convolutional neural network (CNN) with only 0.13 million parameters, enabling processing speeds exceeding 1,000 frames per second (FPS), and up to 2,100 FPS. The method is validated on in vivo calcium, voltage, and volumetric time-lapse imaging data from mice and zebrafish. Key contributions include the spatiotemporal sampling strategy, the lightweight architecture, remarkable processing speed, and a graphical user interface (GUI) for integration into imaging workflows. The authors demonstrate improved neuron segmentation, preservation of rapid voltage transients, and enhanced 3D calcium event extraction.

Strengths:

- 1) The reported processing speeds are indeed impressive, particularly for applications like closed-loop experiments
- 2) The use of a 0.13M parameter model is noteworthy for achieving high performance, minimizing computational overhead and memory requirements.
- 3) The validation across three imaging modalities and different model organisms demonstrates the potential versatility of FAST.
- 4) The self-supervised nature, leveraging spatiotemporal redundancies and avoiding the need for clean ground truth data for training, is highly practical for many experimental setups.

Response:

We greatly appreciate the reviewer's positive evaluation of our work. We have carefully revised the manuscript to address all the concerns and improve the clarify of the manuscript.

Comment 1:

The authors attribute the success of their 0.13M parameter model to the “meticulously designed ultra-lightweight 2D U-Net”, specific architectural choices, and the “spatiotemporal balanced training strategy”. They also state that their “spatial subsampler can be regarded as a single downsampling step that expands the receptive field, thereby allowing us to reduce the number of downsampling steps in the U-Net to shrink the network.”

While these explanations are provided, the link between these specific choices and the exceptional performance on diverse, complex biological data could be strengthened. An ablation study demonstrating the impact of each key architectural modification (e.g., with/without group convolutions, with/without the top-level skip connection, varying downsampling depths) on both performance and speed would provide more convincing evidence for why this specific lightweight design works so well.

The Discussion mentions that “complexity is unnecessary.” While this is a strong claim, a more nuanced discussion comparing the trade-offs between model complexity, receptive field, and the nature of noise/signal in fluorescence imaging would be beneficial.

Response:

Based on your suggestions, we have addressed these mentioned points and expanded the discussion accordingly:

1. Quantitative analysis of architectural choices via ablation studies

As suggested, we have performed a set of ablation studies to evaluate the contribution of each key architectural component to our model’s performance and efficiency. These results are presented in Supplementary Section 4 and Supplementary Figure 4. We also expanded the discussion accordingly in the revised manuscript (Page, 20, Line 483). Herein, we summarized the key findings below:

- **Group convolutions:** a model variant using standard convolutions instead of group convolutions was tested. It resulted in a > 25-fold increase in parameters and a concurrent decrease in the Signal-to-Noise Ratio (SNR). This result highlights the effectiveness of group convolutions in achieving parameter efficiency in this application.
- **Top-level skip connection:** our baseline model does not include a top-level skip connection. We tested a variant with addition of this connection. It led to a marginal increase in SNR but also nearly doubled the total parameter counts. This result supports our design choice to prioritize model compactness and inference speed for practical deployment scenarios.
- **Down-sampling depth:** our experiments indicated that a depth of three down-sampling steps provides an effective trade-off. Reducing the depth to two led to a noticeable drop in SNR, while increasing the depth to four offered a negligible SNR gain relative to the increase in model size and computational cost.
- **Channel width:** we tested alternative channel widths to our chosen value of 64. Increasing the width to 128 improved SNR but also more than doubled the parameter counts. Decreasing the width to 32 resulted in significantly lower performance. This analysis supports 64 channels as a suitable choice for balancing performance and resource utilization in our model.

In summary, these ablation studies provided quantitative data that explain our architectural choices. The findings indicated that each component contributes to achieving a favorable balance between denoising performance, model size, and computational load for the target biological imaging data.

2. A more nuanced discussion on model complexity

In the revised manuscript, we have expanded our analysis and discussion as regard to the interplay between model complexity, receptive field, and the specific noise/signal characteristics inherent to fluorescence imaging.

- Regarding to “model complexity”: we stated that, while increased model complexity (e.g., more layers, wider channels, additional skip connections) can theoretically enhance representational capacity, our empirical results indicate diminishing returns for the fluorescence microscopy data considered in this study. Specifically, the ablation results show that, beyond a certain level, further increases in complexity do not yield proportional improvements in denoising performance, but do incur significant computational and memory costs.
- Regarding to “receptive field”: we clarified that, our spatial subsampler, by effectively expanding the receptive field with fewer down-sampling steps, enables the model to capture relevant contextual information without excessive depth. This design is particularly well-suited to the spatial statistics of noise and signal in fluorescence images, where local context is critical for denoising.
- Regarding to “the specific noise/signal characteristics inherent to fluorescence imaging”: we highlighted that the noise in fluorescence microscopy is often spatially and temporally structured, rather than purely random. Our model’s architecture, especially the spatiotemporal balanced training strategy, is tailored to exploit these characteristics, enabling robust denoising with fewer parameters.
- We explicitly acknowledge that for other imaging modalities or tasks with more complex noise profiles, higher model complexity or specialized architectural designs may be necessary.

We have removed the previous statement in Discussion and replaced it with a more balanced one: “We note that for other imaging modalities or tasks characterized by more complex noise patterns, higher model complexity or specialized architectural designs may be required to achieve optimal denoising performance.”

Page 20, Line 483:

“FAST, with its spatiotemporal balanced training strategy, achieves denoising speeds exceeding 1,000 FPS on a single GPU using only a lightweight 2D convolutional network. Previous denoising methods for fluorescence time-lapse imaging often relied on large-scale network structures, such as 3D convolutional networks, Transformers, or combinations of multiple architectures^{15,16,21,24}. While these complex models have strong representational capacity, our results suggest that, within the scope of this study, such complexity does not bring significant advantages. Our ablation studies show that increasing model complexity leads to only marginal improvements in denoising performance, while substantially increasing computational and memory costs. The flexibility of FAST is rooted in two

key tunable parameters: the temporal window width (C) and the temporal shift step (S). These parameters provide users with precise control over the method’s performance based on their specific data and application needs. The parameter S primarily governs the trade-off between processing speed and denoising fidelity. As our analysis shows (Fig. 1b), increasing S dramatically boosts processing speed, though this comes at the cost of a moderate decrease in performance metrics like SSIM. This offers a clear choice: a larger S can be employed for real-time applications where throughput is critical, while a smaller S should be selected for offline analyses prioritizing maximum image fidelity. The parameter C controls the temporal context provided to the model. For datasets with very fast signal dynamics, we recommend using a smaller C. This narrows the temporal receptive field, preventing distinct, rapid events from being overly smoothed and thus preserving crucial temporal precision. Therefore, by jointly tuning S and C, users can flexibly adapt FAST to a wide range of imaging scenarios, balancing computational demands with the need to faithfully represent complex signal dynamics. The performance of FAST meets the real-time denoising demands of most imaging systems, while still leaving time windows available for additional post-processing operations, such as cluster analysis of large-scale neuronal imaging, real-time spike inference, functional connectivity analysis, online activity pattern recognition, optogenetic stimulation feedback, and automated ROI selection and tracking⁴⁴⁻⁴⁸. We note that for other imaging modalities or tasks characterized by more complex noise patterns, higher model complexity or specialized architectural designs may be required to achieve optimal denoising performance.”

Supplementary Section 4:

“Supplementary Section 4: Quantitative comparison of different network variants based on ablation experiments.

To systematically assess the impact of key architectural choices, we conducted a series of ablation experiments. Each variant represents a modification to one aspect of the network design, and the results are visualized in terms of denoising performance (SNR), model size, and inference speed.

The data used for these ablation studies were generated using the NAOMI simulator to produce noise-free calcium imaging sequences. As stated in Supplementary Section 3, to simulate realistic imaging conditions, we subsequently added synthetic Poisson-Gaussian noise to the data, ensuring a controlled and reproducible evaluation environment.

The FAST model (our baseline, highlighted in red) achieves a strong balance among all three metrics: it delivers a substantial improvement in SNR (23.27 dB) over the raw input data (11.11 dB, indicated by the dashed line), while maintaining an exceptionally compact parameter count (0.0136M) and fast inference. In contrast, replacing group convolutions with standard convolutions dramatically increases the parameter count (over 25-fold) and reduces SNR, underscoring the effectiveness of group convolutions for both efficiency and performance.

Adding a top-level skip connection yields a marginal SNR gain but nearly doubles the model size, suggesting diminishing returns relative to increased complexity. Varying the number of downsampling steps reveals that reducing depth to two severely compromises performance, while increasing depth to four offers only slight SNR improvement at the cost of a larger model. Similarly, increasing channel width to 128 improves SNR but more than doubles the parameter count, whereas reducing it to 32 channels leads to performance degradation.

Overall, these results demonstrate that the FAST architecture is well-calibrated for the specific demands of biological fluorescence microscopy denoising: it achieves competitive performance with minimal computational and memory requirements. The ablation studies validate our design choices, confirming that further increases in model complexity do not yield proportional benefits for this application scenario.”

Supplementary Figure 4:

“**Supplementary Figure 4: Quantitative comparison of different network variants based on ablation experiments.** The horizontal axis shows the network architecture variants, with the “Baseline (without Top-level Skip Connection, with Group Conv, 3 Downsampling Steps, 64 Channels)”, which is the parameter setting chosen in our main manuscript, highlighted in red. The vertical axis represents the denoising performance measured by SNR (dB) for each model. The SNR of the raw input data (11.11 dB) is indicated by a horizontal dashed line for reference. Each point corresponds to a specific model variant; bubble size is proportional to the number of model parameters (in millions), and color represents processing speed (FPS), mapped using the jet colormap (blue indicates slower speed, red indicates faster speed). Variants include the baseline, a version without group convolutions (Plain Conv), a variant with an added top-level skip connection, models with 2 or 4 downsampling steps, and models using 32 or 128 channels. This visualization summarizes the trade-offs between model complexity, denoising performance, model size, and inference speed.”

Comment 2:

The manuscript compares FAST with DeepCAD-RT, SRDTrans, DeepVid, and SUPPORT. For the calcium imaging dataset, it’s stated that raw videos were “denoised using five different methods... followed by temporal maximum intensity projection (MIP) and segmentation using Cellpose.”

It is crucial to detail how these competitor models were run. Were their pre-trained models used (if available and applicable)? Or were they retrained on the specific datasets presented in this

manuscript? If retrained, how were their hyperparameters selected and optimized? The concern that FAST might be meticulously tuned for the presented datasets while competitor models are not (or run with default/suboptimal parameters) is valid and needs to be explicitly addressed to ensure fair comparison. This is particularly important for self-supervised methods where training is dataset-specific.

The authors should state whether they used the official implementations of the compared methods and what, if any, parameter tuning was performed for them on the datasets used in this study.

Response:

For all four competitor methods (DeepCAD-RT, SRDTrans, DeepVid, and SUPPORT), we utilized the official implementations released by the respective authors. Specifically, the codebases used for each method correspond exactly to the official implementations linked by the Repository URLs provided in Supplementary Table 2. Each of these methods is self-supervised and, as per their original publications, requires model training on the specific dataset to be denoised. We closely followed the recommended training pipelines described in their official documentation to ensure methodological consistency and fairness.

All these details, including any deviations from default settings, are documented in Supplementary Table 2. To avoid any possible oversight, we further emphasize the existence and content of Supplementary Table 2 in the revised manuscript, both in the Methods section and at relevant points in the main text, to ensure that readers can easily find this information.

Supplementary Table 2: List of denoising methods compared in this study.

Method	Repository URL	Framework	Network configuration	Reference
DeepCAD-RT	https://github.com/cabooster/DeepCAD-RT	PyTorch	Changed patch_xy to 50, patch_t to 25 from default	Li, X. et al. Nat Biotechnol 41, 282–292 (2023).
SRDTrans	https://github.com/cabooster/SRDTrans	PyTorch	Changed patch_x to 32, patch_t to 32 from default	Li, X. et al. Nat Comput Sci 3, 1067–1080 (2023).
DeepVid	https://github.com/bu-cisl/DeepVID	TensorFlow	Default	Platisa, J. et al. Nat Methods 20, 1095–1103 (2023).
SUPPORT	https://github.com/NICALab/SUPPORT	PyTorch	Default	Eom, M. et al. Nat Methods 20, 1581–1592 (2023).

Comment 3:

The GCaMP6s data used for evaluation is from Ref. 32, which links to the Neurofinder challenge (neurofinder.codeneuro.org). Neurofinder is a recognized public benchmark. The authors perform their own segmentation and evaluation against manual ground truth. It would strengthen the claims if they could (a) compare their quantitative segmentation results to published/leaderboard results for Neurofinder for the same dataset instance if available, or (b) clearly justify why their specific evaluation protocol is a robust alternative for comparison.

It is more recommended to benchmark on a widely adopted, independent public challenge dataset with a leaderboard.

Response:

The primary objective of Fig. 2 is to systematically evaluate the impact of various denoising methods on neuronal segmentation performance. In the current study, we adopted Cellpose as a unified segmentation model (Stringer et al., 2021). Cellpose is a widely recognized, general-purpose segmentation tool, and importantly, its training data includes two-photon calcium imaging neuronal segmentation tasks, making it well suited for our application scenario. We have also described all details in the Methods section under “Neuron segmentation and evaluation metrics.” In the revised manuscript, we have further elaborated on these points in the main text to enhance clarity and transparency.

We actually evaluated the other mainstream neuronal segmentation methods such as CaImAn and Suite2p on the same dataset. Both of them showed comparable performance with Cellpose, however, they were highly dependent on parameter tuning, making it difficult to ensure a fair and reproducible comparison when assessing the effect of denoising. In contrast, Cellpose requires only a single parameter (target cell diameter), which greatly reduces variability and potential bias. We therefore selected Cellpose as a unified segmentation backend in order to compare the performance among different denoising methods.

We have clarified these points and the rationale for our evaluation protocol in the revised manuscript. We appreciate your comments, which have helped us to further strengthen our work.

Page 26, Line 654:

“For segmentation, we selected Cellpose as a unified model in this study. Cellpose is a general-purpose segmentation tool whose training data includes two-photon calcium imaging neuronal segmentation, making it applicable to our task. Its use of a single adjustable parameter (target cell diameter) also facilitates standardized evaluation across datasets. We note that other supervised segmentation methods may achieve good performance but typically require retraining or fine-tuning on specific datasets, which is not aligned with our goal of an unbiased and consistent evaluation pipeline.”

Comment 4:

The manuscript states: “Data are available from the corresponding author upon reasonable request.” While code is available on GitHub, this data availability statement is a significant barrier to independent reproduction and verification, which is critical for a journal like Nature Communications.

Recommendation: The authors should be strongly encouraged to deposit all raw data used to generate the figures into a publicly accessible repository (e.g., Zenodo, Figshare, a specialized neuroscience data archive) without requiring a request. This would substantially bolster the claims of reproducibility.

Response:

In response to your recommendation, we have deposited all raw datasets utilized in this study in the Zenodo public repository. Specifically, the repository now contains: (1) the in vivo neuronal population calcium imaging dataset used in our experiments, (2) simulated voltage fast dynamics data, (3) single-cell voltage imaging data with corresponding electrophysiological recordings, (4) zebrafish

light-sheet imaging data, and (5) astrocyte volume imaging data. All data are accessible without restriction at <https://doi.org/10.5281/zenodo.15872025>.

We believe this modification addresses your concern by removing barriers to independent reproduction and verification, thereby enhancing the rigor and transparency of our work. If any relevant data have been inadvertently omitted, we are committed to making these available upon notification.

Accordingly, we have updated the Data Availability statement in the manuscript:

Page 33, Line 897:

“Data availability

All raw datasets and relevant materials used in this study are publicly available on Zenodo at <https://doi.org/10.5281/zenodo.15872025>.”

Comment 5:

The authors acknowledge that previous methods have used Transformers. Their argument for not using a more complex model like a Transformer is primarily centered on the desire for an “ultra-lightweight” network to achieve extreme processing speeds, stating in the Discussion, “we argue that such complexity is unnecessary.”

This is a reasonable justification given their primary goal of real-time processing. However, a brief discussion acknowledging if any lightweight Transformer variants were considered and why the CNN approach was ultimately favored might be helpful.

Response:

Thanks for the comments. In the revised manuscript, we have expanded our discussion to clarify both the rationale for our current selection and our perspective on future developments.

Our primary goal of the presented work was to design a framework that could achieve extreme processing speeds for real-time applications. While we evaluated a range of modern efficient architectures, our analysis concluded that a lightweight CNN currently offers the most favorable trade-off for this specific objective. To ensure this rationale is clearly articulated in the manuscript, we have now included a new paragraph in the Discussion. This discussion explicitly states that our choice was driven by the practical advantages of CNNs in speed and efficiency, which were critical for our goal.

Furthermore, we agree it is crucial to acknowledge the rapid evolution of this field. Our revised discussion therefore concludes with a forward-looking perspective, recognizing the potential of emerging technologies and affirming our commitment to future evaluation.

Page 21, Line 523:

“In selecting our network architecture, we also carefully considered a range of modern efficient architectures, including recent lightweight Transformer models^{49,50}. Currently, CNN-based solutions still offer clear advantages in inference speed, computational efficiency, and implementation

simplicity for the extreme real-time and low-latency demands of high-frame-rate applications^{51,52,53}. We therefore adopted a lightweight CNN as the backbone of FAST. Nevertheless, we fully acknowledge the distinct advantages of those advanced efficient models, particularly lightweight Transformers, with their powerful ability to model long-range dependencies and capture global contextual information. This capability is especially promising for tackling complex, non-local noise patterns. As these architectures are continuously optimized for greater computational efficiency, they are poised to become a compelling alternative for real-time denoising.”

References:

- “49. Mehta, S. & Rastegari, M. Mobilevit: Light-weight, general-purpose, and mobile-friendly vision transformer. in International Conference on Learning Representations (2022).
50. Wu, K. et al. Tinyvit: Fast pretraining distillation for small vision transformers. in European conference on computer vision 68 – 85 (Springer, 2022).
51. Berroukham, A., Housni, K. & Lahraichi, M. Vision transformers: A review of architecture, applications, and future directions. in 2023 7th IEEE Congress on Information Science and Technology (CiSt) 205 – 210 (IEEE, 2023). doi:10.1109/CiSt56084.2023.10410015.
52. Han, K. et al. A survey on vision transformer. IEEE transactions on pattern analysis and machine intelligence 45, 87 – 110 (2022).
53. Saha, S. & Xu, L. Vision transformers on the edge: A comprehensive survey of model compression and acceleration strategies. Neurocomputing 643, 130417 (2025).”

Comment 6:

(Remarks on code availability)

The FDU-donglab/FAST repository implements a self-supervised denoising algorithm for 3D image time-series, using a 2D U-Net with a “frame-multiplexing” strategy and a composite loss function. The core FAST algorithm appears to be implemented as described.

Reproducibility of published results is partially supported by aspects like configuration saving and random seed management. However, the lack of specified library versions, exact dataset details, and some hardcoded parameters might require effort to precisely match paper outcomes.

As a community resource, the code’s modular design is a plus. However, it does not have a comprehensive test suite. The lack of comprehensive tests also impacts its robustness and maintainability, which is often a requirement for good software. Addressing these documentation and safety issues would greatly enhance its value to the community.

Response:

In response to your feedback, we have made several improvements to the FDU-donglab/FAST repository (<https://github.com/FDU-donglab/FAST>).

We have now specified all required library versions and provided a more detailed environment configuration guide, including both pip and conda environment files. Additionally, we have included

information about the operating systems (both standalone and server), GPU models, and CUDA versions that we have tested, to further support reproducibility.

To enhance user experience, we have not only retained the concise Quick Start commands, but also added a comprehensive Usage Guide. This guide describes, in detail, how to train the model on custom datasets in three steps and how to use trained models for data processing in two steps, along with other important notes and troubleshooting tips. Furthermore, random seed management and configuration saving are now clearly documented, and previously hardcoded parameters have been minimized or clarified. To foster community engagement, we have enabled the GitHub Discussions feature, allowing users to participate in open discussions and collaborative problem-solving directly within the repository.

We believe these updates address the concerns you raised regarding documentation, environment specification, and community support, and we are committed to further improving the robustness and maintainability of the codebase in future releases.

During the revision process, we identified and corrected a minor error regarding the parameter count of our proposed method. In the original submission, it was mistakenly reported as 0.13M, while the correct value is 0.013M. This correction further highlights the lightweight nature of our network. The manuscript has been updated accordingly.

Page 4, Line77:

“Here, we present the FrAme-multiplexed SpatioTemporal learning strategy (FAST), enabling real-time, self-supervised denoising across diverse neural imaging scenarios. FAST is carried out within an ultra-lightweight 2D convolutional network containing only **0.013M** parameters.”

Page 24, Line 614:

“By minimizing the network depth, the number of channels per feature map, and the number of skip connections, our network trains with only approximately **0.013M** parameters.”

Reviewer #2:

The manuscript entitled “Real-time self-supervised denoising for high-speed fluorescence neural imaging” by Wang et al. described a real-time image denoising technique for functional neuronal imaging. The authors demonstrated the method on multiple calcium and voltage imaging datasets. This is an interesting work, and I appreciate the primary novelty of the technique’s ability to perform real-time denoising.

Response:

We greatly appreciate the reviewer’s positive evaluation of our work. We have carefully revised the manuscript to address all the concerns and improve the clarity of the manuscript.

Comment 1:

From the reported demonstrations, I fail to see the necessity of real-time denoising, as all results were obtained post hoc. It would strengthen the paper if the authors could include at least one experiment that directly benefits from real-time processing.

Response:

Our post-hoc benchmarks demonstrate that our method’s accuracy is on par with, and in some cases superior to, computationally intensive offline algorithms. Its key distinction, however, is the ability to deliver these results concurrently with data acquisition. It is important to clarify that even when using offline datasets for these benchmarks, our method operated in a simulated real-time mode, processing each frame sequentially as if from a live data stream. This ensures that our performance metrics are representative of a true real-time application.

To explicitly demonstrate this real-time capability and its necessity, as you requested, we have performed an additional *in vivo* experiment. This experiment shows our method facilitating live neuroscience research with immediate, actionable feedback, a process detailed in the newly added Supplementary Section 8 and Supplementary Video 1.

The practical value of this robust real-time capability is substantial, opening avenues for more sophisticated experimental designs:

- Immediate Guidance for Discovery: Researchers can instantly identify ROIs with weak signals, allowing for dynamic experimental adjustments to capture critical neural events.
- Mitigating Phototoxicity for Long-Term Imaging: With a clear signal visible in real time, researchers can confidently use lower laser power, enabling stable, long-duration imaging.
- Enabling Advanced Closed-Loop Experiments: The low-latency feedback is a critical component for future closed-loop systems that rely on instantaneous brain activity.

Building upon the real-time denoising capability established in this work, future development can focus on integrating downstream analysis modules (such as signal extraction and event detection) to create a fully integrated, on-the-fly processing workflow.

In summary, while our offline benchmarks validate our method’s accuracy, its core contribution is advancing the experimental paradigm from purely post-hoc analysis towards interactive, real-time

guidance. This is made possible by our method's exceptional efficiency, which makes powerful denoising tools more accessible for day-to-day neuroscience research.

Supplementary Section 8:

“Supplementary Section 8: Real-time denoising of two-photon calcium imaging in mice with FAST.

We used Matlab-based ScanImage software for image acquisition. Before starting the experiment, imaging control parameters were preset. FAST was then launched, and the data read and save paths, model path, and buffer size were configured. The FAST system consists of two main components: a Matlab-based application and a Python-based deep learning denoising module. Acquired image data is buffered and accessed in real time by the Python module for denoising and visualization. During the experiment, both the raw and denoised images are displayed simultaneously, enabling real-time comparison. After processing, the denoised data is automatically saved to disk. The video demonstrates an acquisition rate of 30 Hz at 512×512 pixels and is shown at real speed. Post-production editing of the screenshot was limited to spatial adjustments (cropping and repositioning) for improved visual presentation, with no modification to original content.”

Supplementary Video 1:

Supplementary Video 1: Real-time denoising of two-photon calcium imaging in mice with FAST.

Comment 2:

Related to point 1, all Results sections are titled “Real-time ... with FAST”. In my understanding, real-time implies that denoising is performed concurrently with image acquisition. However, the demonstrated experiments are not the case. The authors may consider describing their method as fast.

Response:

Our original use of “Real-time” in the section titles was intended to describe the nature of our processing algorithm itself. We would like to clarify that even when applied to offline data, our method operates in a simulated real-time manner, processing each frame sequentially, thus mimicking a live data stream. In response to your suggestion, we have revised the manuscript to ensure precise and unambiguous terminology. To avoid confusion between our method name (FAST) and the adjective “fast,” and to more accurately describe our processing approach, we have replaced “real-time” in all relevant section titles and subheadings with “rapid.”

- “Real-time intracellular Ca²⁺ imaging of neuronal populations with FAST” is now “**Rapid processing of intracellular Ca²⁺ imaging data from neuronal populations with FAST.**”
- “Real-time enhancement in high-speed voltage imaging with FAST” is now “**Rapid enhancement in high-speed voltage imaging with FAST.**”
- “Real-time processing of volumetric imaging of cortical astrocytes with FAST” is now “**Rapid processing of volumetric imaging of cortical astrocytes with FAST.**”

We believe this revision improves both the clarity and accuracy of our presentation, while still reflecting the high-speed capability of our method.

Comment 3:

Comments on Fig. 2. In Fig. 2h, the blue circular overlays are visually distracting. For example, it is difficult to assess whether FAST performs better on Cells 2 and 3. For Cell 2, all methods seem to produce discontinuous morphology by eye. For Cell 3, some discontinuous circle-like structures are also visible with SRDTrans and DeepVid, but the blue circle on FAST makes it hard to visually compare the results. Therefore unfortunately, I am not convinced that “..., for Cell 2, only SRDTrans and FAST enabled accurate segmentation, with other methods either over-smoothing the boundaries or introducing artifacts. Notably, only FAST successfully segmented Cell 3, recovering subtle neuronal structures obscured by noise.” The authors may consider using arrowheads instead. The authors also claimed that “FAST-denoised images revealed synaptic structures ...”, can the authors use arrows to indicate the synaptic structures? From the current images, they are not clearly visible. For Fig. 2i, how many neurons were used? In addition, SUPPORT produced some horizontal fringes – we have tested SUPPORT on our calcium datasets and didn’t observe such artifacts. Could the authors comment on this?

Response:

In Figure 2, we compared the effects of five denoising methods on calcium imaging data by applying each method prior to neuronal segmentation with Cellpose. Our responses to your specific comments are as follows:

1. Regarding the distracting overlays in Fig. 2h and claims on Cells 2 & 3:

- **Action on overlays:** We have replaced the circular overlays with arrowheads in the revised Figure 2h.
- **Action on claims:** We have moderated the language in the manuscript to state that FAST facilitates more accurate segmentation, rather than claiming it as an absolute outcome. The revised text now details the comparison as follows (Page 10, Line 232): “We further compared the raw and denoised images in Fig. 2h, focusing on the segmentation outcomes for three representative neurons (Cells 1–3) to illustrate the challenges of segmentation under noisy conditions. For Cell 1, all methods enabled successful segmentation by Cellpose, as its contour remained relatively clear and intact across the different denoising results. For Cell 2, only the SRDTrans- and FAST-denoised images allowed Cellpose to segment the neuron, whereas the other methods produced results with insufficiently continuous structures, making segmentation difficult. For Cell 3, the FAST-denoised image exhibited the most complete contour and the clearest boundary, which greatly facilitated accurate segmentation.”

2. Regarding the visibility of structures in Fig. 2g:

- Action on annotations & terminology:** As you suggested, we have added red arrowheads to the magnified view and revised the term “synaptic structures” to the more accurate term “dendrites”: “In the magnified view of Fig. 2g, arrowheads indicate dendrites that were originally obscured by noise in the raw data but became discernible after denoising with FAST.”

3. Regarding the number of neurons for the analysis in Fig. 2i:

- Action:** We have specified in the figure legend that the analysis was performed on a total of **155 neurons**. “i, We compared the segmentation performance for 155 neurons using raw images versus those from five denoising methods, based on Accuracy, Recall, and F1 score.”

4. Regarding the horizontal fringe artifacts from the SUPPORT method:

- Our comments:** We hypothesize the artifacts are due to a data mismatch. SUPPORT was optimized for high-speed voltage imaging, and its performance is likely sensitive to differences in data acquisition rates. In our case, the method was applied to calcium imaging data acquired at a comparatively lower frame rate (8 Hz), which may partly explain the

observed artifacts. This may also explain the discrepancy compared to the experience with potentially faster datasets. A note on this point has been added to the manuscript. “We also observed that the SUPPORT method produced some horizontal fringe artifacts in our calcium imaging dataset, which may be related to differences in imaging modality and acquisition rate compared to its original application in voltage imaging.”

Revisions to the manuscript include:

Figure 2:

Fig. 2 | FAST enhances Ca^{2+} imaging quality and analytical accuracy. a, Raw GCaMP6s calcium imaging videos from the mouse vS1 region were first denoised using different denoising methods.

Subsequently, temporal maximum intensity projection (MIP) was applied to the videos. The resulting projection images were then segmented using Cellpose. **b**, In vivo neural population calcium imaging data were obtained from Ref. 32. The top panel shows the raw calcium imaging data with a red box indicating a region of interest (ROI). Enlarged images of the ROI at four different time points during a single spike event (~7.5 s time window) are presented, illustrating neuronal activity in various states to evaluate denoising performance. The bottom panel displays the corresponding segmentation results, with manual annotations used as ground truth: correctly segmented regions (true positives) are shown in green, missed regions (false negatives) in red, and extra regions (false positives) in blue. Three neuron regions are highlighted with boxes in the bottom panel, and their corresponding magnified segmentation results are shown in **h**. Scale bar, 100 μm . **c-g**, Example denoising and segmentation results from raw data processed using DeepCAD-RT, SRDTrans, DeepVid, SUPPORT, and FAST. Following the same structure as **b**. In **g**, arrowheads indicate the fine dendrites that become observable after FAST denoising. **h**, Example neurons, zoomed in from the boxed regions in **b-g**, illustrating segmentation results across different denoising methods. Each row represents a single neuron, while each column shows the maximum intensity projection of the raw data or data denoised by one of the five methods. In **h**, asterisks indicate neurons that were successfully segmented by Cellpose. Scale bar, 5 μm . **i**, We compared the segmentation performance for 155 neurons using raw images versus those from five denoising methods, based on Accuracy, Recall, and F1 score.

Page 10, Line 228:

“After denoising, FAST demonstrated superior performance in restoring neuronal morphology and improving segmentation outcomes compared to other methods (Figs. 2c–g, Supplementary Section 9 and Supplementary Video 2). In the magnified view of Fig. 2g, arrowheads indicate dendrites that were originally obscured by noise in the raw data but became discernible after denoising with FAST. We further compared the raw and denoised images in Fig. 2h, focusing on the segmentation outcomes for three representative neurons (Cells 1–3) to illustrate the challenges of segmentation under noisy conditions. For Cell 1, all methods enabled successful segmentation by Cellpose, as its contour remained relatively clear and intact across the different denoising results. For Cell 2, only the SRDTrans- and FAST-denoised images allowed Cellpose to segment the neuron, whereas the other methods produced results with insufficiently continuous structures, making segmentation difficult. For Cell 3, the FAST-denoised image exhibited the most complete contour and the clearest boundary, which greatly facilitated accurate segmentation.”

Page 11, Line 247:

“We also observed that the SUPPORT method produced some horizontal fringe artifacts in our calcium imaging dataset, which may be related to differences in imaging modality and acquisition rate compared to its original application in voltage imaging.”

Comment 4:

Comments on Fig. 4. In Fig. 4b, I am not sure whether the improved image quality is primarily due to denoising. I tried background subtraction by (1) removing a constant-value background and (2) using a 100/200-pixel-diameter rolling ball, and obtained images that appeared similar to those from FAST.

Although I didn't do detailed analysis, the authors may want to further verify that the improvement is not due to basic background removal. The overlaid colors are also distracting. The authors should at least show side-by-side comparisons of raw and denoised images without overlays, and provide a color bar indicating activity strength. In the caption, some words need further clarification: "extracted events were fragmented, incomplete, and often missing. After ... more continuous, complete, and accurately defined". It is also not clear what the effective volume rate is. From Methods, the frame rate is 30Hz, and the data were acquired with $2 \times 512 \times 512 \times 25 \times 336$: does that mean the volume rate is $30/25=1.2$ Hz? Is this sufficient to capture GCaMP6s-reported calcium activities? Finally, in Methods: "The imaging volume was ... achieving a lateral resolution of 0.2 μm and an axial resolution of 2.4 μm ." This is not accurate – resolution is determined by the NA of the objective lens, sampling, aberration, and other factors.

Response:

1. Regarding the source of image quality enhancement in Figure 4b:

We have revised Fig. 4b to present side-by-side comparisons of raw and denoised images at four representative time points, using an identical color bar for both sets. This direct comparison clearly demonstrates that the observed enhancement with FAST arises from the denoising process itself, rather than from simple background subtraction.

The apparent differences in background between the raw and denoised images are primarily attributable to shot noise, which is particularly pronounced in low-signal regions under low signal-to-noise acquisition conditions. FAST effectively suppresses this background noise, resulting in a cleaner appearance that may superficially resemble the outcome of background removal. However, we confirm this improvement is achieved entirely through denoising, without any explicit background subtraction step.

2. Regarding the terminology in the Figure 4 caption:

To address the need for clarification on the terms "fragmented," "incomplete," and "missing" events, we have now provided precise definitions for each term in the revised caption:

- **Fragmented events:** This refers to the phenomenon where a single, continuous signal is erroneously divided into multiple smaller events during detection.
- **Incomplete events:** This describes detected events that fail to fully cover the true spatiotemporal extent of the underlying signal.
- **Missing events:** This refers to real signal events that are not detected at all.

3. Regarding the effective volume rate and its sufficiency:

Your calculation is correct; the effective volume an imaging rate of 1.2 Hz. Our primary objective with this specific experiment was not to capture all possible calcium transients, but rather to characterize the volumetric dynamics across the entire soma of individual astrocytes. This 3D, single-cell approach provides unique insights into slow, whole-cell signaling events, which is a distinct and valuable area of investigation.

We acknowledge that this imaging rate represents a necessary trade-off. To achieve this volumetric coverage with our current setup, we sacrificed temporal resolution. Consequently, while

this rate is well-suited for resolving the large-scale somatic events that are the focus of this analysis, it will inevitably miss some of the faster, localized calcium activities, particularly in the finer processes.

Addressing this limitation is a key direction for our ongoing research. We are actively exploring and developing advanced imaging techniques to achieve higher temporal resolution in volumetric imaging in our future work.

4. Regarding the description of “resolution” in the Methods:

You are correct that our description was inaccurate. True optical resolution is determined by multiple factors beyond sampling. Our intention was to state the voxel size of our imaging system. We have revised the Methods section to clarify that the values of 0.2 μm (lateral) and 2.4 μm (axial) represent the voxel size, and we have removed the term “resolution” in this context to avoid ambiguity.

Revisions to the manuscript include:

Figure 4:

Fig. 4 | FAST improves astrocytic Ca^{2+} events quantification in real-time spatiotemporal volumetric imaging. **a**, Two-photon imaging of cortical astrocytes was performed in awake mice. Cortical astrocytes were virus-labeled with GCaMP6s (denoting Ca^{2+} signals, in green) and tdTomato (denoting morphology, in red) fluorescence channels. **b**, Direct comparison of raw and FAST-denoised images at representative time points. Images from the $z = 15$ plane are displayed for four selected frames (3.7 s, 7.4 s, 181.3 s, and 351.5 s), with the top row showing the raw data and the bottom row showing the corresponding FAST-denoised results. All images are displayed using an

identical intensity scale, and a unified color bar indicating signal intensity is provided in the lower left corner of the first panel. This side-by-side comparison illustrates that FAST denoising effectively suppresses noise and enhances the visibility of astrocytic Ca²⁺ signals, particularly in regions with low signal-to-noise ratio. Scale bar, 20 μm . **c**, Comparison of the raw data and data processed with FAST. The first row shows the raw data, and the second row shows the FAST-denoised data. The first column on the left displays the volumetric view of the calcium imaging channel. The second to fourth columns show images of the $z=15$ slice at three representative frames (7.4 s, 181.3 s and 351.5 s), with overlaid colors indicating calcium activity events in the current frame, extracted using the AQUA method³⁹. Different events are distinguished by different colors. In the raw data, extracted events were fragmented (a single continuous event erroneously detected as multiple separate segments), incomplete (detected events failing to capture the full spatiotemporal extent of the underlying signal), and often missing (true events not detected at all). After FAST denoising, the extracted events appeared more continuous, complete, and accurately defined, exhibiting spatiotemporal characteristics that align more closely with human perception. Scale bar, 20 μm . **d-f**, Quantification of event features extracted from the $z=15$ plane in b for RAW and FAST data. The bar plots show the comparison of event area (**d**), event perimeter (**e**), and event circularity (**f**). Statistical significance was assessed using the Kolmogorov-Smirnov test ($***P < 0.001$). Data are presented as mean \pm SEM. **g**, The spatiotemporal distribution of Ca²⁺ events within the astrocytic territory. It displays the spatiotemporal distribution map of Ca²⁺ events extracted based on raw data (left) and FAST-processed data (right). Each circle represents a Ca²⁺ event, with the center coordinates corresponding to the event's spatial center and the diameter (unit, μm) scaled according to the event's volume. The color scale denotes the duration of individual Ca²⁺ events (ranging from 0-5 s).

Page 28, Line 718:

“**Imaging protocols:** Imaging was performed using a two-photon microscope with resonant scanning at 30 Hz, where the objective lens was axially scanned to capture 3D volumes. The imaging volume was scanned at a resolution of 512 \times 512 pixels across 25 layers with a 2.4 μm increment in the z-axis, achieving a lateral voxel size of 0.2 μm , an axial voxel size of 2.4 μm , and an effective volume rate of 1.2 Hz.”

Page 18, Line 429:

“In the raw data, noise obscured many events and diminished the clarity of their spatial and temporal features.”

Minor comments:

1. Line 427: “structrual” should be “structural”.

Response: We have corrected the typo in the revised manuscript.

2. In line 677: the sentence “there is no correspondence between the same color across different images” is unclear.

Response:

We have revised the sentence to clarify that colors are randomly assigned to events on a per-frame basis; therefore, within a single image (frame), the same color represents the same event, but the same color in different frames does not indicate correspondence between events across time.

Page 29, Line 729:

“**Event extraction for a single plane:** Astrocyte calcium events were extracted from individual planes using the AQuA software analysis package³⁹. Masks were created in AQuA to exclude regions outside the cells, the minimum size of the connected region was set to 4, the intensity threshold scaling factor was set to 7, and the standard deviation of the Gaussian filter was set to 0.2. To enhance image detail, the event area masks were overlaid onto the images in Matlab. **In this visualization, colors are randomly assigned to events on a per-frame basis; thus, within a single frame, the same color represents the same event, but the same color in different frames does not indicate correspondence between events across time.**”

3. The dataset from Ref 35 was acquired using SIM. The authors may want to clarify that it refers to background-rejection SIM, not super-resolution SIM.

Response:

We have carefully reviewed Ref. 35 and agree that clarification is warranted. In the referenced work, structured illumination microscopy (SIM) was primarily employed for in vivo voltage imaging to reduce out-of-focus fluorescence and improve signal-to-noise ratio (SNR), rather than to achieve super-resolution. As described in the context of NoRMCorre’s residual motion artifacts, the use of structured illumination significantly enhanced SNR in fixed samples by minimizing out-of-focus fluorescence, highlighting the background-rejection capability of this SIM implementation.

We have revised our description to explicitly state that “background-rejection structured illumination fluorescence microscopy” was used, to avoid any confusion with super-resolution SIM.

Page 12, Line 292:

“**Imaging was conducted at 1,000 Hz using background-rejection structured illumination fluorescence microscopy³⁵, alongside simultaneous patch-clamp electrophysiological recordings.**”

Reviewer 3:

The manuscript introduces FAST, a deep-learning method for real-time denoising in fluorescence microscopy, optimized through a compact neural network and a paired-frame, self-supervised training approach. FAST achieves high processing speeds (>1000 FPS) and preserves rapid neural signals, representing a valuable technical improvement over existing techniques. While the work is practically significant for high-speed neural imaging, its incremental advancement relative to previously published methods such as DeepCAD-RT (Li et al., Nature Biotechnology, 2023) limits conceptual novelty. The analysis presented is sound, but the interpretation should explicitly acknowledge limitations, particularly regarding sensitivity to motion artifacts and blood flow. Critically, the manuscript primarily shows post-processed examples rather than explicitly demonstrating the method's real-time capabilities, such as closed-loop applications or live imaging.

Comment 1:

Technical Novelty: The manuscript represents an incremental advancement over existing methods such as DeepCAD-RT, primarily through enhanced computational efficiency and processing speed. To strengthen the manuscript, authors should explicitly highlight unique conceptual differences compared to prior methodologies.

Response:

In response, we have revised the manuscript to more explicitly highlight the unique conceptual differences of our approach compared to prior methodologies, particularly DeepCAD-RT.

1. Core conceptual difference: paradigm shift in information utilization

Our work introduces a fundamental shift from the traditional “simple sampling, complex network” paradigm to a “smart sampling, simple network” approach. Specifically:

2. Previous methods (e.g., DeepCAD-RT):

- Rely on temporally adjacent frames as training pairs, assuming signal quasi-staticity.
- Employ complex 3D network architectures (e.g., 3D U-Net with ~ 1.0 M parameters) to learn spatiotemporal correlations.
- Are computationally intensive and limited when dealing with rapid dynamic signals.

3. Our approach (FAST):

- Proposes a novel frame-multiplexed spatiotemporal sampling strategy that intelligently integrates spatial and temporal neighbors into training pairs.
- Enables the use of an ultra-lightweight 2D network (0.013M parameters, only 1.2% of DeepCAD-RT), entirely bypassing the need for 3D convolutions.
- This transition from a 3D to a 2D paradigm is made possible solely by our unique sampling method, not by incremental architectural tweaks.

The dramatic improvements in speed and computational efficiency are not merely optimizations but natural consequences of this paradigm shift. By eliminating the need for heavy 3D convolutions, our method achieves real-time processing while simultaneously enhancing signal fidelity in

challenging experimental regimes. These conceptual differences are summarized and illustrated in Figure 1.

4. Overcoming dynamic signal distortion through principled flexibility

Prior methods, such as DeepCAD-RT, relying on temporal redundancy are limited to quasi-static signals and perform poorly in ultra-fast imaging scenarios (e.g., voltage imaging), as shown in Figure 3 a-c. These methods often misinterpret rapid signal changes as noise, causing over-smoothing and loss of critical information.

In contrast, FAST addresses this limitation through its inherent flexibility, rooted in two key tunable parameters: the temporal window width (C) and the temporal shift step (S). These provide users with precise control over the method’s performance based on their specific data and application needs:

- **The parameter S** governs the trade-off between processing speed and denoising fidelity. As our analysis shows (Fig. 1b), increasing S dramatically boosts speed at the cost of a moderate decrease in performance metrics like SSIM. This offers a clear choice: a larger S for real-time applications where throughput is critical, and a smaller S for offline analyses prioritizing maximum image fidelity.
- **The parameter C** controls the temporal context provided to the model. For datasets with very fast signal dynamics, we recommend using a smaller C. This narrows the temporal receptive field, preventing distinct, rapid events from being overly smoothed and thus preserving crucial temporal precision.

This ability to jointly tune S and C is a key conceptual advantage, allowing FAST to be adapted to diverse imaging scenarios—a level of user-controlled flexibility not offered by prior, more rigid methodologies.

In summary, FAST represents a non-incremental advance in real-time image denoising, defined by a novel sampling paradigm that enables both a radically different network architecture and access to experimental domains previously inaccessible to conventional methods.

We have further revised the Introduction and Methods to explicitly articulate the conceptual innovation underpinning FAST. We have also clarified the limitations of previous approaches in the Discussion. We believe that these clarifications and revisions better communicate the conceptual novelty and significance of our work.

Revisions to the manuscript include:

Page 4, Line 77:

“Here, we present the FrAme-multiplexed SpatioTemporal learning strategy (FAST), enabling real-time, self-supervised denoising across diverse neural imaging scenarios. FAST is carried out within an ultra-lightweight 2D convolutional network containing only **0.013M** parameters. This lightweight architecture significantly enhances computational efficiency, enabling real-time processing even on resource-limited hardware. The model processes full-size, multi-frame images without partitioning, breaking through the denoising speed ceiling of 1,000 frames per second (FPS) and achieving an unprecedented speed of up to 2,100 FPS. **This approach differs from existing**

real-time denoising methods, which typically depend on fixed temporal sampling and complex 3D network architectures to capture spatiotemporal information. FAST, by introducing an adaptive frame-multiplexed spatiotemporal sampling strategy, enables the use of a lightweight 2D network without sacrificing denoising performance. This design allows the temporal window to be flexibly adjusted according to the signal dynamics, facilitating a tailored balance between spatial and temporal information. Consequently, FAST can effectively reduce artifacts and over-smoothing, particularly in scenarios involving rapid or non-stationary neural activity. By decoupling the reliance on heavy network structures and instead leveraging more informative sampling, FAST extend the applicability of real-time self-supervised denoising to experimental scenarios that have previously posed significant challenges for conventional approaches.”

Page 19, Line 459:

“Noise presents a significant challenge in intravital neural observation, with inherent shot noise setting the ceiling for the SNR in fluorescence imaging, thereby constraining resolution, speed, and sensitivity. Effective denoising is crucial, and different imaging scenarios require tailored approaches: structural imaging benefits from high spatial resolution, while functional imaging demands high temporal resolution. Self-supervised deep learning methods have emerged as powerful tools for denoising, and leveraging spatiotemporal redundancy in raw data. However, existing approaches often involve complex training and resource-intensive processes, and it remains challenging to provide the fast, high-quality denoising techniques needed for neuroscience research to support real-time data analysis and closed-loop neural modulation. Unlike previous approaches that primarily focus on increasing network complexity to extract spatiotemporal correlations, FAST introduces a paradigm shift in information utilization by employing a novel frame-multiplexed spatiotemporal sampling strategy. This shift enables the use of a much simpler 2D network architecture, fundamentally distinguishing FAST from prior methods. We introduce FAST, a universal real-time self-supervised denoising pipeline for diverse neural imaging scenarios. FAST uses a lightweight 2D convolutional network to balance temporal and spatial redundancies, achieving rapid denoising speeds beyond 1,000 FPS, and extreme to 2,100 FPS. It has been validated across various imaging scenarios and subjects, demonstrating superior denoising performance and real-time capability. FAST enhances cellular morphology restoration, neuron segmentation, and 3D calcium event extraction while preserving spike shapes and improving voltage transient correlations with electrophysiological recordings. A user-friendly GUI further facilitates training, offline inference, and real-time online inference, enabling state-of-the-art denoising of user data.”

Page 20, Line 483:

“Previous denoising methods for fluorescence time-lapse imaging often relied on large-scale network structures, such as 3D convolutional networks, Transformers, or combinations of multiple architectures^{15,16,21,24}. While these complex models have strong representational capacity, our results suggest that, within the scope of this study, such complexity does not bring significant advantages for fluorescence microscopy denoising. FAST, with its spatiotemporal balanced training strategy, achieves denoising speeds exceeding 1,000 FPS on a single GPU using only a lightweight 2D convolutional network. It is important to note that the feasibility of using an ultra-lightweight 2D CNN is not achieved by incremental architectural tweaks, but rather is made possible by our unique

FAST sampling strategy, which intelligently balances spatial and temporal information during training. Our ablation studies show that increasing model complexity leads to only marginal improvements in denoising performance, while substantially increasing computational and memory costs. The flexibility of FAST is rooted in two key tunable parameters: the temporal window width (C) and the temporal shift step (S). These parameters provide users with precise control over the method's performance based on their specific data and application needs. The parameter S primarily governs the trade-off between processing speed and denoising fidelity. As our analysis shows (Fig. 1b), increasing S dramatically boosts processing speed, though this comes at the cost of a moderate decrease in performance metrics like SSIM. This offers a clear choice: a larger S can be employed for real-time applications where throughput is critical, while a smaller S should be selected for offline analyses prioritizing maximum image fidelity. The parameter C controls the temporal context provided to the model. For datasets with very fast signal dynamics, we recommend using a smaller C. This narrows the temporal receptive field, preventing distinct, rapid events from being overly smoothed and thus preserving crucial temporal precision. Therefore, by jointly tuning S and C, users can flexibly adapt FAST to a wide range of imaging scenarios, balancing computational demands with the need to faithfully represent complex signal dynamics. The performance of FAST meets the real-time denoising demands of most imaging systems, while still leaving time windows available for additional post-processing operations, such as cluster analysis of large-scale neuronal imaging, real-time spike inference, functional connectivity analysis, online activity pattern recognition, optogenetic stimulation feedback, and automated ROI selection and tracking^{44–48}. We note that for other imaging modalities or tasks characterized by more complex noise patterns, higher model complexity or specialized architectural designs may be required to achieve optimal denoising performance.”

Page 22, Line 534:

“Existing self-supervised time-lapse denoising methods rely on temporal redundancy to recover true signals from noisy data. These methods are based on the assumption that information carriers maintain ergodicity across adjacent temporal coordinates. This foundational assumption, however, confronts inherent limitations when the temporal sampling rate is low or the signal's dynamics are fast. Hence, in practical ultra-high speed imaging acquisition, such as neuronal voltage imaging, the dynamics between a spike window are too sharp to maintain the assumption after temporal subsampling, and further lead deviations that can lead to distortion and smoothed spatial structure of recovered signals. Most current methods seek to maximize the temporal input range, aiming to enhance denoising performance by capturing potential signal features. However, a wider temporal input range often contains several dynamics. A self-supervised approach relying solely on temporal redundancy would cause a phantasm of temporal causality, wherein signal restoration highly depends on future signals, especially in 3D convolutional networks. This temporal causality phantasm inevitably introduces uncertainties in ultra-high speed imaging, compromising the authenticity of the restored signal. **This conceptual advance allows FAST to flexibly adapt to diverse signal dynamics by decoupling the reliance on strict temporal continuity, thus overcoming the limitations of traditional self-supervised frameworks in handling rapid or non-ergodic neural events.** FAST mitigates this issue by incorporating adjustable trade-off parameters that constrain the temporal receptive field and expand the spatial receptive field. By tuning the information-receiving range, FAST effectively

minimizes interference at its source, preserving the authenticity of neural signals while reducing noise. This controlled approach enables the rapid and faithful restoration of high-fidelity signals without relying on extensive spatiotemporal inputs.”

Comment 2:

Real-time Processing Demonstration: Although the manuscript emphasizes FAST’s high processing speeds as its key contribution, it predominantly presents post-processed results. Explicit demonstrations or quantitative evaluations of real-time performance (e.g., latency measurements, closed-loop imaging scenarios) are essential to convincingly justify its need.

Response:

Our manuscript predominantly featured post-processed results for a crucial reason: to enable a fair and rigorous comparison against other state-of-the-art denoising methods, which are typically applied offline. It is also important to clarify that even when processing these offline datasets, our method operated in a simulated real-time mode, processing each frame sequentially, to ensure our performance metrics are representative of a true real-time application. The fundamental pipeline and design for this real-time processing were detailed in Figure 1e and 1f of our original manuscript.

To provide the explicit real-time demonstrations and quantitative evaluations that you requested, we have now built upon this foundation with several key additions, including a systematic latency analysis and a live demonstration video.

We have added Supplementary Video 1 (detailed in Supp. Section 7) to demonstrate real-time processing during a live in vivo two-photon calcium imaging session. This video shows raw and denoised images side-by-side during a 30 Hz acquisition, illustrating the seamless data transfer, denoising, and visualization enabled by our integrated Matlab-Python framework.

The total system latency is composed of two main components:

- **Initialization latency:** The time required to process the initial batch of frames. This is intrinsic to online systems and can be minimized by reducing the initial batch size.
- **Processing latency:** The time to process each subsequent frame. On our hardware (NVIDIA RTX 3090), the processing speed for 512×512 images consistently exceed the 30 Hz acquisition rate, meaning processing latency is not a limiting factor in our setup (see Fig. 1d, Fig. S2 for benchmarks).

Relevant descriptions have been incorporated into the revised manuscript.

Page 8, Line 181:

“The system’s real-time performance and responsiveness are further demonstrated in Supplementary Video 1 and Supplementary Section 8, where a live in vivo two-photon calcium imaging experiment is presented. In this demonstration, both raw and denoised images are displayed simultaneously at an acquisition rate of 30 Hz, providing direct visual evidence of FAST’s capability for real-time denoising under practical experimental conditions.”

Supplementary Section 8:

Supplementary Section 8: Real-time denoising of two-photon calcium imaging in mice with FAST.

“We used Matlab-based ScanImage software for image acquisition. Before starting the experiment, imaging control parameters were preset. FAST was then launched, and the data read and save paths, model path, and buffer size were configured. The FAST system consists of two main components: a Matlab-based application and a Python-based deep learning denoising module. Acquired image data is buffered and accessed in real time by the Python module for denoising and visualization. During the experiment, both the raw and denoised images are displayed simultaneously, enabling real-time comparison. After processing, the denoised data is automatically saved to disk. The video demonstrates an acquisition rate of 30 Hz at 512×512 pixels and is shown at real speed. Post-production editing of the screenshot was limited to spatial adjustments (cropping and repositioning) for improved visual presentation, with no modification to original content.”

Supplementary Video 1:

Supplementary Video 1: Real-time denoising of two-photon calcium imaging in mice with FAST.

Comment 3:

Limitations in handling motion and broadband noise: FAST fundamentally relies on minimal motion between frames. Substantial motion and broadband noise such as bloodflow would significantly decrease its efficacy. The manuscript should address and acknowledge this limitation and state the necessity of incorporating image registration or motion correction as fast as this processing technique.

Response:

In response, we have conducted additional analyses to systematically evaluate FAST’s performance under challenging conditions, specifically regarding motion artifacts and broadband noise, and have accordingly expanded our discussion on the method’s practical limitations and application boundaries.

To systematically evaluate the robustness of FAST to motion artifacts, we performed simulations with varying amplitudes of frame-to-frame displacement, as described in the newly added Supplementary Section 13. The results, summarized in Supplementary Figure 11, show that FAST maintains high denoising performance under no or slow motion, but its effectiveness declines noticeably under vigorous motion. We fully acknowledge this limitation of FAST when substantial motion is present, and emphasize that, in practical applications, FAST should be combined with suitable image registration or motion correction techniques as a preprocessing step. These findings and recommendations are now explicitly discussed throughout the main text and supplementary materials.

To further evaluate FAST’s performance on data with broadband noise, we tested its denoising performance on blood flow imaging data (Fig. R1). The data used here were directly obtained from Meng et al. [PNAS 119, e2117346119 (2022)], where 1 kHz 2D full-frame imaging of capillaries at a depth of 560 μm below the dura in awake mouse cortex was performed using a free-space angular chirp enhanced delay (FACED) two-photon fluorescence microscope (2PFM) with FITC-dextran

labeling (field of view: $50\ \mu\text{m} \times 100\ \mu\text{m}$). As shown in Fig. R1, FAST can significantly improve SNR while maintaining good spatial correspondence with the original images. While these results demonstrate the potential of FAST for denoising data with significant broadband noise, we recognize that further optimization may be needed for such scenarios. To this end, we are actively exploring dedicated denoising methods tailored for broadband noise, and a more comprehensive analysis will be provided in future work.

Fig. R1. FAST denoising on 1 kHz two-photon blood flow imaging. A single red blood cell is tracked over a period of 28 ms, with its trajectory indicated by green arrows. The first row displays the raw images, and the second row shows the corresponding FAST-denoised images. The time interval between the first five frames is 1 ms, and for the last five frames it is 5 ms. The scale bar represents $10\ \mu\text{m}$.

We have revised the relevant sections to clearly acknowledge the limitations of FAST in the presence of substantial motion and broadband noise, and to stress the importance of integrating rapid motion correction procedures in practical imaging scenarios. We hope these additions address your concerns and provide a transparent discussion of the method’s applicability and boundaries.

Page 21, Line 516:

“To further clarify the boundaries of FAST, we systematically evaluated its robustness to motion artifacts and broadband noise. As detailed in Supplementary Section 13 and Supplementary Figure 11, FAST maintains high denoising performance under no or slow motion, but its effectiveness declines with vigorous motion. Therefore, in practical applications involving substantial motion, we recommend combining FAST with image registration or motion correction as a preprocessing step.”

Supplementary Section 13:

“Supplementary Section 13: Evaluating the impact of motion artifacts on FAST denoising performance.

To systematically evaluate the robustness of FAST to different degrees of motion artifacts, we generated simulated two-photon calcium imaging data using NAOMI and introduced mixed Poisson-Gaussian noise to mimic realistic imaging conditions. To simulate different motion levels, we introduced a random displacement between adjacent frames, with the magnitude (in pixels) varying

across seven conditions. Seven motion conditions were tested: no motion (0), slow motion (0–1, 0–2, 0–3, 0–4, 0–5), and vigorous motion (4–5).

Each dataset was processed with FAST for denoising, and the structural similarity index (SSIM) was calculated by comparing the denoised images with the ground truth. As shown in Supplementary Figure 11, FAST achieved consistently high SSIM values under no motion and all slow motion conditions (SSIM = 0.59 for 0 and 0–1, 0.58 for 0–2 and 0–3, 0.57 for 0–4 and 0–5), indicating robust denoising performance despite minor motion. In contrast, the SSIM dropped substantially to 0.46 in the vigorous motion group (4–5), demonstrating that large inter-frame displacements can significantly impair denoising quality. For reference, the SSIM of the noisy raw images was 0.21 (dashed line in the figure). These results highlight the importance of motion stabilization for optimal denoising with FAST, especially under conditions of vigorous motion.”

Supplementary Figure 11:

Supplementary Figure 11: Evaluating the impact of motion artifacts on FAST denoising performance. Simulated two-photon calcium imaging data were generated using NAOMI, with mixed Poisson-Gaussian noise artificially added. Seven groups of data were generated by introducing random rigid motion between adjacent frames with varying distance ranges per step (in pixels): no motion (0), slow motion (0–1, 0–2, 0–3, 0–4, 0–5), and vigorous motion (4–5). Each dataset was denoised using FAST, and the structural similarity index (SSIM) between the denoised results and the ground truth was calculated. The x-axis indicates the magnitude of simulated motion, defined as the range of random displacement (in pixels) introduced between adjacent frames for each condition. The y-axis shows the corresponding SSIM values (mean ± SD). The dashed line indicates the SSIM (0.21) of the noisy raw images. FAST maintained high SSIM values under no motion and slow motion conditions (0: 0.59, 0–1: 0.59, 0–2: 0.58, 0–3: 0.58, 0–4: 0.57, 0–5: 0.57), but showed a marked drop in performance under vigorous motion (4–5: 0.46).

Comment 4:

Robustness and Generalizability: Authors should clarify the range of conditions (noise levels, indicator brightness, neuron types) under which FAST reliably performs. A thorough exploration of robustness would significantly strengthen claims of applicability.

Response:

Our manuscript demonstrates the robustness and generalizability of FAST across a wide range of imaging platforms, biological models, and experimental conditions. Specifically, we have validated its performance on:

- Microscopy Platforms: Data acquired from various common imaging systems, including confocal, light-sheet, and multi-photon microscopes.
- Fluorescent Indicators: A variety of widely-used probes, such as the calcium sensor GCaMP6s and the voltage sensors QuasAr6a and zArchon1.
- Biological Models: Different cell types (neuronal and astrocytic) from both mice and zebrafish.
- Simulated Conditions: A broad spectrum of noise levels, imaging speeds, and signal-to-noise ratios, as detailed in our systematic simulations (Supplementary Section 1, Fig. S1).

This multi-faceted evidence, combining diverse real-world data and systematic simulations, provides a thorough demonstration of FAST's broad applicability.

Comment 5:

Parameter Sensitivity and Trade-offs: The manuscript should address the sensitivity of FAST's performance to key parameters such as frame interval selection and spatial-temporal weighting, outlining potential trade-offs.

Response:

We would like to first clarify that our method does not involve "spatial-temporal weighting." The key parameters governing the trade-off between performance and speed are the temporal window width (C) and the temporal shift step (S).

As detailed in our original submission, we had already addressed the sensitivity and trade-offs associated with these parameters in several places:

- In the Methods section (describing the framework in Fig. 1a), we explicitly introduced S and C and stated that they "balance temporal resolution and processing speed, with larger strides S increasing speed."
- In the Results section, we provided a quantitative evaluation of this trade-off. Figure 1b and its accompanying text demonstrated how increasing the stride S significantly boosts processing speed to over 1,000 Hz while maintaining high denoising performance (PSNR and SSIM).

However, we agree that this is a critical point and that a more consolidated discussion would benefit the reader. Therefore, to make these trade-offs more prominent and accessible, we have now added a dedicated summary of these analyses to the Discussion section in the revised manuscript.

Page 20, Line 494:

“The flexibility of FAST is rooted in two key tunable parameters: the temporal window width (C) and the temporal shift step (S). These parameters provide users with precise control over the method’s performance based on their specific data and application needs. The parameter S primarily governs the trade-off between processing speed and denoising fidelity. As our analysis shows (Fig. 1b), increasing S dramatically boosts processing speed, though this comes at the cost of a moderate decrease in performance metrics like SSIM. This offers a clear choice: a larger S can be employed for real-time applications where throughput is critical, while a smaller S should be selected for offline analyses prioritizing maximum image fidelity. The parameter C controls the temporal context provided to the model. For datasets with very fast signal dynamics, we recommend using a smaller C. This narrows the temporal receptive field, preventing distinct, rapid events from being overly smoothed and thus preserving crucial temporal precision. Therefore, by jointly tuning S and C, users can flexibly adapt FAST to a wide range of imaging scenarios, balancing computational demands with the need to faithfully represent complex signal dynamics.”

Comment 6:

Biological Interpretability: The manuscript provides limited insight into how improved technical image quality meaningfully enhances biological interpretability. Demonstrating explicit biological or analytical advantages would strengthen the practical relevance and impact.

Response:

To address how FAST’s technical improvements enhance biological interpretability, we focus on the new experimental capabilities unlocked by its real-time denoising performance. Several cutting-edge biological experiments critically depend on the ability to interpret neural activity on-the-fly, a need that FAST is specifically designed to meet:

1. Enabling closed-loop neuromodulation:

Many advanced experimental designs require real-time feedback, such as delivering a stimulus precisely when a specific pattern of neural activity is detected. A raw, noisy signal makes this impossible. By providing a clean, low-latency data stream, FAST makes these closed-loop experiments practical, allowing for real-time interrogation and manipulation of neural circuits.

2. Real-time guidance for in vivo experiments:

During live imaging, researchers must make critical decisions based on immediate visual feedback. With a continuously denoised view, an experimenter can:

- **Use significantly lower laser power**, as weak signals become clearly visible. This minimizes phototoxicity and enables much longer, more stable imaging sessions.
- **Identify and target specific, weakly active cells** for intervention (e.g., for patching or targeted photostimulation) that would otherwise be invisible in the noise.

3. Immediate assessment of experimental manipulations:

As shown in our astrocyte imaging results (Fig. 4), FAST allows for the immediate and more accurate extraction of calcium events. This provides an instant readout of cellular activity, enabling researchers to assess the impact of a drug application or behavioral stimulus in real time and adjust the experimental protocol accordingly.

In essence, the primary biological advantage of FAST is not merely to refine post-hoc analysis, but to shift the experimental paradigm from passive data collection towards interactive, real-time investigation, where biological interpretation and intervention can happen concurrently.

One minor comment:

Voltage imaging is typically not analyzed per pixel basis, as a practical note, signals are extracted from ROI as pixel averages. Moreover, the randomly selected pixels should be taken from the cell membrane and not the cytoplasm due to the way the sensor labels neurons.

Response:

We fully acknowledge your point that, in typical voltage imaging analyses, signals are extracted as pixel averages from membrane regions due to the membrane localization of the sensor. In our study, we deliberately chose to randomly sample pixels from the entire neuronal ROI, including both membrane and cytoplasmic regions. This approach provides a more rigorous and comprehensive evaluation of denoising performance, as cytoplasmic pixels generally have lower signal-to-noise ratios. By including both compartments, we set a higher bar for method robustness and demonstrate its applicability under realistic and challenging imaging conditions. We have revised the relevant section of the manuscript to clarify this methodological choice.

Page 12, Line 290:

“To validate FAST’s performance on real-world data, we applied it to in vivo voltage imaging of single neurons expressing QuasAr6a in layer 2/3 of the mouse cortex (Fig. 3d, Supplementary Section 10 and Supplementary Video 3). Imaging was conducted at 1,000 Hz using background-rejection structured illumination fluorescence microscopy³⁵, alongside simultaneous patch-clamp electrophysiological recordings. The raw imaging data were heavily contaminated with noise, which obscured neuronal boundaries and distorted voltage signals. **FAST denoising significantly improved the clarity of neuronal morphology and restored voltage signals, as demonstrated by the $\Delta F/F$ traces extracted from 1000 pixels randomly selected from the entire neuronal ROI, including both membrane and cytoplasmic regions (Fig. 3d, right panel).** In the enlarged time window shown in Fig. 3e, FAST-denoised traces closely matched the electrophysiological recordings, accurately capturing the timing and amplitude of individual spikes. To quantitatively evaluate FAST’s performance, we calculated the Pearson correlation coefficients between the $\Delta F/F$ traces and the simultaneously recorded electrophysiological signals (Fig. 3f). FAST significantly increased the correlation coefficients compared to the raw data ($P < 0.001$), indicating a substantial improvement in signal fidelity.”

Revised Fig. 3:

Fig. 3 | High-fidelity voltage imaging enhancement via FAST. **a**, Workflow of the simulation experiment to evaluate denoising performance on fast dynamic signals. Two-photon fluorescence neural imaging data were simulated, with each neuron assigned distinct virtual spike signals of varying widths (2 ms, 4 ms, 6 ms, or 8 ms), generating noise-free ground truth (GT) data containing only neuronal signals. Mixed Poisson-Gaussian (MPG) noise was then added to create noisy data, which served as input for denoising networks to evaluate their performance. **b**, Statistical analysis of denoising performance across spike widths. Pearson correlation coefficients ($\Delta F/F$) were calculated between each neuron's raw or denoised signals (using five different methods) and the ground truth signal across four spike widths (2 ms, 4 ms, 6 ms, 8 ms). Data are presented as violin plots with individual data points. Statistical significance was determined using two-way ANOVA followed by Tukey's multiple comparisons test ($n=10$, representing the number of neurons). Significance levels are indicated as follows: NS ($P \geq 0.05$), ** ($P < 0.01$, decrease), *** ($P < 0.001$, decrease), ## ($P < 0.01$, increase), ### ($P < 0.001$, increase). **c**, Comparison of $\Delta F/F$ traces around spikes extracted from denoised images across different methods and spike widths. Representative $\Delta F/F$ traces extracted

from a single neuron are shown, aligned to spike events. From top to bottom: ground truth trace, noisy trace (Noisy), and traces extracted from images denoised by DeepCAD-RT, SRDTrans, DeepVid, SUPPORT, and FAST. From left to right: traces corresponding to spike widths of 2 ms, 4 ms, 6 ms, and 8 ms. **d**, Raw and FAST-denoised images with in vivo simultaneous voltage imaging and electrophysiology. Left: Raw and FAST-denoised images of QuasAr6a-expressing pyramidal neurons in the mouse cortex (L2/3). Imaging was performed in vivo using simultaneous structured illumination fluorescence imaging and patch-clamp electrophysiological recordings at a frame rate of 1,000 Hz (The data were obtained from Ref. 35). Right: The top trace shows the patch-clamp electrophysiological recording obtained simultaneously with imaging. **The middle and bottom panels show $\Delta F/F$ traces extracted from 1000 randomly selected pixels within the neuron ROI, which was annotated using the ROI Manager toolbox in Fiji³⁶.** The middle panel represents traces from noisy data, while the bottom panel shows traces from FAST-denoised data, with each row corresponding to a single pixel. A red dashed box highlights a specific time window, which is shown at higher magnification in panel **e**. Scale bar, 10 μm . **e**, Enlarged view of the time window highlighted by the red dashed box in **d**. From top to bottom, the panel shows the patch-clamp electrophysiological trace, followed by $\Delta F/F$ traces of three representative pixels from the raw and FAST-denoised data. For direct comparison, the raw and FAST-denoised traces are overlaid. Red triangles and gray dashed lines indicate the peak and corresponding timing of spikes in the electrophysiological trace. **f**, Pearson correlation coefficients between the 100 single-pixel $\Delta F/F$ traces and the patch-clamp electrophysiological recordings shown in **a**. The correlation significantly improves after FAST denoising (** $P < 0.001$, unpaired two-tailed t-test). Data are presented as mean \pm SEM. **g**, Voltage imaging of zebrafish spinal cord neurons using light sheet microscopy. Imaging was performed on the neural population in the zebrafish spinal cord using the voltage probe zArchon1 at a frame rate of 1000 Hz. The data were obtained from Ref. 15. The raw and FAST-denoised images are shown, with neuron ROIs manually annotated. Three representative cells are highlighted by boxes, and their enlarged views are displayed on the right for comparison. Scale bar, 20 μm (left), 5 μm (right). **h**, $\Delta F/F$ traces of three neurons extracted from **g**. From left to right, the traces represent RAW and FAST-denoised data, and from top to bottom, the traces correspond to Cell 1, Cell 2, and Cell 3. A red dashed box highlights a specific time window, which is shown in an enlarged view in **i**. **i**, Enlarged view of the time window highlighted in **h**. To facilitate comparison, the RAW and FAST-denoised traces are overlaid.

Point-by-point response to the reviewers' comments

Title: Real-time self-supervised denoising for high-speed fluorescence neural imaging

Author: *Yiqun Wang, Yuanjie Gu, Jianping Wang, Ang Xuan, Cihang Kong, Wei-Qun Fang, Dongyu Li, Dan Zhu, Fengfei Ding, and Biqin Dong*

We greatly appreciate the reviewers' positive evaluation of our work and their constructive comments and suggestions. We have carefully revised the manuscript to address all the concerns and improve the clarity of the manuscript. We highlight the revised sections with the red color in the revised manuscript. Please see our point-by-point response to reviewers' comments below:

Reviewer #1:

(Remarks to the Author)

The authors have thoroughly addressed my concerns raised in the previous round of review. The manuscript has been significantly strengthened through the inclusion of new supporting data, expanded analyses, and important clarifications throughout the text. The authors' commitment to transparency and reproducibility is commendable, and the revised manuscript is a much more compelling and robust piece of work. I would recommend it for publication.

(Remarks on code availability)

The code appears to be of high quality. The documentation is clear, and the setup is straightforward, though I didn't install and run the code.

Response:

We are sincerely grateful for your positive assessment of our revised manuscript and code. Your constructive feedback from the first round was instrumental in enhancing the work, and we truly appreciate your support for our work.

Reviewer #2:

(Remarks to the Author)

The authors performed additional experiments and provided clarifications to some of my earlier concerns. Below are my remaining comments.

Comment 1:

(Comments on the necessity of doing real-time)

The authors conducted real-time denoising with an in vivo mouse brain imaging experiment. It demonstrated the speed advantage, but it didn't show how real-time denoising benefits biological studies. I agree with what the authors listed; these are, of course, practical values of real-time denoising, but the direct benefit was not demonstrated.

Response:

The primary contribution of this work is the introduction and demonstration of the technical capability and real-time performance of the FAST pipeline. Our intent is to provide an enabling technology that lays the groundwork for the biological applications you mentioned. As we explicitly note in the Discussion: "The performance of FAST meets the real-time denoising demands of most imaging systems, while still leaving time windows available for additional post-processing operations, such as cluster analysis of large-scale neuronal imaging, real-time spike inference, functional connectivity analysis, online activity pattern recognition, optogenetic stimulation feedback, and automated ROI selection and tracking."

We hope this clarification sufficiently addresses your concern. In addition, we are actively planning future studies that will apply FAST in real-time experimental paradigms to directly demonstrate its utility in advancing biological discovery.

Comment 2:

(Comments on Figure 2)

This part includes 155 neurons demonstrating the segmentation performance and 3 neurons showing cell morphology and segmentation accuracy.

Overall, the current data do not support the statement, "FAST demonstrated superior performance in restoring neuronal morphology and improving segmentation outcomes compared to other methods." For the statistical analysis in Fig. 2i – All methods are mostly comparable except for SUPPORT – differences are really subtle.

For the three selected cells, given the non-significant statistical differences shown in Fig 2i, the most I can conclude is that in some cases, FAST works better. In addition, it looks like all the images in Fig. 2h were individually normalized (from 0/min to max, the authors may want to confirm this), except for the image with DeepCAD-RT, which was obviously much dimmer. The authors should check how images were normalized for fair comparisons.

Also, the marked dendrites can also be visualized using DeepCAD-RT and SRDTrans.

Lastly, I am not convinced by the explanation of why SUPPORT produced fringes – in the original paper, the authors (Eom et al. 2023, Nature Methods) already demonstrated calcium imaging applications. The lower frame rate should not be the reason.

Response:

1. On the comparison of performance (Fig. 2i) and the statement of "superior

performance”:

Figure 2 shows that FAST’s denoising quality and segmentation outcomes are statistically comparable to those of state-of-the-art, computationally intensive offline methods. This is an important result, as it demonstrates that the substantial gains in efficiency and speed provided by our ultra-lightweight model do not come at the expense of performance. The three example neurons in Fig. 2h highlight challenging cases where subtle differences in denoising can determine the success or failure of downstream automated processes such as Cellpose segmentation. To better reflect this balance, we have revised the manuscript language to emphasize that FAST delivers competitive performance.

Page 7, Line 166:

“After denoising, FAST demonstrated excellent performance in restoring neuronal morphology and improving segmentation outcomes (Figs. 2c–g, Supplementary Section 9 and Supplementary Video 2).”

2. On image normalization for fair comparison (Fig. 2h):

We would like to clarify that no individual normalization was applied to the magnified views. These panels were directly cropped from globally normalized images without any further re-normalization. The lower brightness observed in the DeepCAD-RT result therefore reflects the true output of that algorithm on this dataset, rather than an artifact of our visualization method. To avoid any ambiguity, we have explicitly added a description of the normalization procedure to the figure caption.

Figure 2’s caption:

“h, Example neurons, zoomed in from the boxed regions in b-g, illustrating segmentation results across different denoising methods. Each row represents a single neuron, while each column shows the maximum intensity projection of the raw data or data denoised by one of the five methods. To ensure a fair visual comparison, all magnified views shown were cropped from full-sized images that had been globally normalized to a single, unified intensity range. In h, asterisks indicate neurons that were successfully segmented by Cellpose. Scale bar, 5 μm .”

3. On the visibility of dendrites:

Our statement is based on visual evidence showing that dendrites in FAST-denoised images exhibit greater clarity and structural continuity compared to other methods. This enhanced visibility is important for reliable morphological analysis. We have revised the manuscript to more precisely indicate that FAST facilitates clearer and more discernible visualization of fine dendritic structures, directly reflecting the results presented.

Page 7, Line 168:

“In the magnified view of Fig. 2g, arrowheads indicate dendrites that were originally obscured by noise in the raw data. After denoising by FAST, these structures were revealed with notably improved clarity and continuity.”

4. On the artifacts produced by the SUPPORT method:

We believe the most constructive path forward is to ground the discussion in transparency and reproducibility. To this end, we have provided all necessary details for independent verification: Supplementary Table 2 lists the official code repositories and the exact parameters used for the SUPPORT implementation, and all raw imaging data used in this comparison are publicly available via our Zenodo repository (<https://doi.org/10.5281/zenodo.15872025>).

While the exact cause of the observed artifacts on this dataset remains speculative, the results shown in the figure are fully reproducible. A detailed analysis of the internal mechanics of a third-party algorithm is beyond the scope of this work. Accordingly, we have added a statement in the revised manuscript reporting the observed, reproducible outcome from SUPPORT without further conjecture.

Page 7, Line 182:

“SUPPORT, despite achieving the highest Accuracy (0.93), showed lower Recall and F1 scores, reflecting a trade-off between precision and recall. **Notably, when applying the official SUPPORT implementation to our calcium imaging dataset, we observed the presence of horizontal fringe artifacts.** FAST, by comparison, balanced these metrics better, achieving a similar Accuracy (0.92) with higher Recall and F1 scores.”

Reviewer #3:

(Remarks to the Author)

The authors have adequately addressed several points in the revised manuscript, and it is much improved. While the authors have done a good job of addressing the Reviewer comments, I would like to point out an issue.

Comment 1:

The abstract mentions “... Utilizing an ultra-light convolutional neural network, FAST enables real-time processing at speeds exceeding 1,000 frames per second...”, but the revision shows calcium imaging data denoising at 30 Hz. Unless real online denoising of fast functional data is shown, I suggest reconsidering the wording of the initial claim, since it might appear confusing to the reader.

Response:

The “>1,000 FPS” figure reflects the benchmarked processing speed of our algorithm, a critical standard for computational performance. For in vivo demonstrations, we used 30 Hz acquisition, as calcium imaging is currently the most common in neural imaging. Under this setting, processing time of FAST is typically <1–2 ms for a 512×512 frame, leaving substantial headroom for downstream operations. While kHz-level throughput is unsuitable for real-time visualization (as it exceeds human perceptual limits), we validated its feasibility through an offline demonstration, as shown in Figure 3.

To prevent misinterpretation, we have rephrased the line in the abstract to explicitly emphasize that our processing throughput far exceeds typical imaging acquisition rates.

Page 2, Line 27:

“Utilizing an ultra-light convolutional neural network, FAST enables processing speeds exceeding 1,000 frames per second, substantially surpassing the acquisition rates of most high-speed imaging systems.”

Comment 2:

Moreover, I would suggest adding a “latency budget” with reasonable values for achieving true live denoising. It can include data readout, motion correction, registration, and the denoising algorithm. I believe having a reasonable expected number would be useful for people using all these methods.

Response:

This is an excellent and highly practical suggestion. We fully agree that for real-world applications, the latency of the entire processing pipeline is a critical consideration. Inspired by your suggestion, we have added additional explanation that outlines the main components of a typical real-time pipeline and introduces the concept of a total latency budget. Importantly, we emphasize that FAST’s own processing time (typically <1–2 ms for a 512×512 frame) represents only a small fraction of the time available between frames (e.g., ~33 ms for a 30 Hz acquisition), leaving a generous margin for other essential—and often more computationally demanding—steps.

We thank you for this constructive input, which has helped us better contextualize the practical utility of FAST for the community.

Page 21, Line 505:

“System latency and pipeline integration

Achieving true real-time performance requires considering the entire pipeline—including data acquisition, transfer, motion correction, and other downstream steps—whose latencies can be substantial and hardware-dependent. A key advantage of FAST is that its processing time represents only a small fraction of this budget (e.g., <1–2 ms for a 512×512 frame versus ~33 ms for a 30 Hz acquisition), leaving ample headroom for more demanding operations and ensuring it does not become a bottleneck in diverse real-time systems.”